# De novo identification of universal cell mechanics gene signatures

Marta Urbanska[1,2]*[†‡], Yan Ge[1†§], Maria Winzi[1], Shada Abuhattum[1,2], Syed Shafat Ali[3,4], Maik Herbig[1,2,5], Martin Kräter[1,2], Nicole Toepfner[1,6], Joanne Durgan[7], Oliver Florey[7], Martina Dori[5], Federico Calegari[5], Fidel-Nicolás Lolo[8], Miguel Ángel del Pozo[8], Anna Taubenberger[1], Carlo Vittorio Cannistraci[1,3,9,10]*, Jochen Guck[1,2]*

[1]Biotechnology Center, Center for Molecular and Cellular Bioengineering, Technische Universität Dresden, Dresden, Germany; [2]Max Planck Institute for the Science of Light & Max-Planck-Zentrum für Physik und Medizin, Erlangen, Germany; [3]Center for Complex Network Intelligence, Tsinghua Laboratory of Brain and Intelligence, Department of Computer Science and School of Biomedical Engineering, Tsinghua University, Beijing, China; [4]Department of Computer Science and Department of Economics, Jamia Millia Islamia, New Delhi, India; [5]Center for Regenerative Therapies Dresden, Center for Molecular and Cellular Bioengineering, Technische Universität Dresden, Dresden, Germany; [6]Klinik und Poliklinik für Kinder- und Jugendmedizin, Universitätsklinikum Carl Gustav Carus, Technische Universität Dresden, Dresden, Germany; [7]Signalling Programme, The Babraham Institute, Cambridge, United Kingdom; [8]Mechanoadaptation and Caveolae Biology Lab, Cell and Developmental Biology Area, Centro Nacional de Investigaciones Cardiovasculares (CNIC), Madrid, Spain; [9]Center for Systems Biology Dresden, Dresden, Germany; [10]Cluster of Excellence Physics of Life, Technische Universität Dresden, Dresden, Germany

*For correspondence:
mu272@cam.ac.uk (MU);
kalokagathos.agon@gmail.com
(CVC);
jochen.guck@mpl.mpg.de (JG)

[†]These authors contributed equally to this work

Present address: [‡]Department of Physiology, Development and Neuroscience, University of Cambridge, Cambridge, United Kingdom; [§]AliveX Biotech, Shanghai, China

## eLife Assessment

This **important** study uses machine learning-based network analysis on transcriptomic data from different tissue cell types to identify a small set of conserved (pan-tissue) genes associated with changes in cell mechanics. The new method, which provides a new type of approach for mechano-biology, is accessible, **compelling**, and well-validated using in silico and experimental approaches. The study provides motivation for researchers to test hypotheses concerning the identified five-gene network, and the method will be strengthened over time with expanded sets of validations, such as testing genes with hitherto unknown roles and different perturbation techniques.

**Abstract** Cell mechanical properties determine many physiological functions, such as cell fate specification, migration, or circulation through vasculature. Identifying factors that govern the mechanical properties is therefore a subject of great interest. Here, we present a mechanomics approach for establishing links between single-cell mechanical phenotype changes and the genes involved in driving them. We combine mechanical characterization of cells across a variety of mouse and human systems with machine learning-based discriminative network analysis of associated transcriptomic profiles to infer a conserved network module of five genes with putative roles in cell mechanics regulation. We validate in silico that the identified gene markers are universal, trust-worthy, and specific to the mechanical phenotype across the studied mouse and human systems, and demonstrate experimentally that a selected target, *CAV1*, changes the mechanical phenotype of cells accordingly when silenced or overexpressed. Our data-driven approach paves the way

toward engineering cell mechanical properties on demand to explore their impact on physiological and pathological cell functions.

## Introduction

The extent to which cells can be deformed by external loads is determined by their mechanical properties, such as cell stiffness. Since the mechanical phenotype of cells has been shown to reflect functional cell changes, it is now well established as a sensitive label-free biophysical marker of cell state in health and disease (*Guck and Chilvers, 2013*; *Nematbakhsh and Lim, 2015*). Beyond being a passive property that can be correlated with cell state, cell stiffness is increasingly recognized as an important feature involved in processes such as development (*Lecuit and Lenne, 2007*; *Hannezo and Heisenberg, 2019*) and cancer progression (*Suresh, 2007*; *Gensbittel et al., 2021*). Identifying the molecular targets for on-demand tuning of mechanical properties is, thus, essential for exploring the precise impact that cell mechanics has on physiological and pathological processes in living organisms.

The mechanical properties of cells are determined by various intracellular structures and their dynamics, with cytoskeletal networks at the forefront (*Fletcher and Mullins, 2010*). According to current knowledge, the most prominent contributor to the global mechanical phenotype is the actin cortex and its contractility regulated via Rho signaling (*Chugh and Paluch, 2018*; *Kelkar et al., 2020*). Intermediate filaments, including vimentin and keratin, reside deeper inside the cell and can also contribute to measured cell stiffness, especially at high strains (*Seltmann et al., 2013*; *Patteson et al., 2020*). Although there is some evidence of the contribution of microtubules to cell stiffness at high strains (*Kubitschke et al., 2017*), their role has been difficult to address directly, since drug-induced microtubule disassembly evokes reinforcement of actin cytoskeleton and cell contractility (*Chang et al., 2008*). Apart from cytoskeletal contributions, the cell mechanical phenotype can be influenced by the level of intracellular packing (*Zhou et al., 2009*; *Guo et al., 2017*) or mechanical properties of organelles occupying the cell interior, such as the cell nucleus (*Caille et al., 2002*). When aiming at modulating the mechanical properties of cells, it may not be practical to target cytoskeletal structures, which are central to a multitude of cellular processes, because their disruption is generally toxic to cells. It is therefore important to identify targets that enable subtle, alternative ways of intervening with cell stiffness.

Most of our knowledge about the molecular contributors to cell mechanics has been derived from drug perturbations or genetic modifications targeting structures known a priori. The challenge of identifying novel targets determining the mechanical phenotype can be addressed on a large scale by performing screens using RNA interference (RNAi) (*Chugh et al., 2017*; *Toyoda et al., 2017*; *Rosendahl et al., 2018*) or small-molecule compound libraries. Alternatively, the problem can be reverse-engineered, in that omics datasets for systems with known mechanical phenotype changes are used for prediction of genes involved in the regulation of mechanical phenotype in a mechanomics approach. Broadly speaking, mechanomics is a study of omics data within the context of mechanobiology. So far, this term has been used with regard to changes in omics profiles in response to an external mechanical stimulus such as shear flow, tensile stretch, or mechanical compression (*Wang et al., 2014*; *Putra et al., 2019*; *Zhang et al., 2021*), or to collectively name all of the mechanical forces acting on or within cells (*National Academy of Engineering, 2008*; *van Loon, 2009*; *Song et al., 2012*; *Song et al., 2013*; *Wang et al., 2021*). However, it can also be used to address omics changes related to changes in the mechanical properties of cells (*Ciucci et al., 2017*; *Poser et al., 2019*) — a context much closer to our study.

Here, we extend the concept of mechanomics to a data-driven methodology for de novo identification of genes associated with the mechanical phenotype based on omics data (*Figure 1*). To demonstrate this approach, we perform a machine learning-based discriminative network analysis termed PC-corr (*Ciucci et al., 2017*) on transcriptomics data from two unrelated biological systems with known mechanical phenotype changes (*Poser et al., 2019*; *Urbanska et al., 2017*) and elucidate a conserved functional module of five candidate genes putatively involved in the regulation of cell mechanics. We then test the ability of each gene to classify cell states according to cell stiffness in silico on six further transcriptomic datasets and show that the individual genes, as well as their compression into a combinatorial marker, are universally, specifically, and trustworthily associated with the mechanical phenotype across the studied mouse and human systems. Finally, we confirm experimentally that one

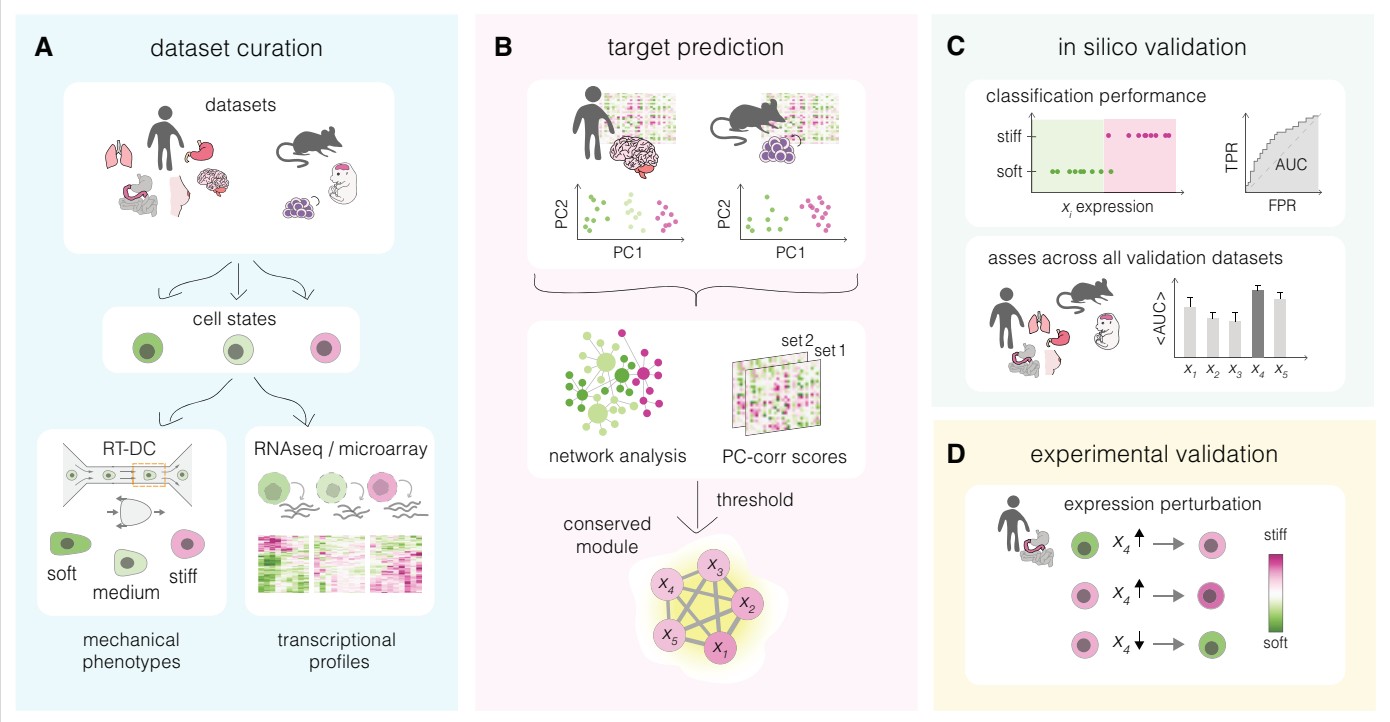

**Figure 1.** Overview of a mechanomics approach for de novo identification of genes involved in cell mechanics regulation. (**A**) Data curation. Datasets originating from different biological systems encompassing cell states with distinct mechanical phenotypes, as characterized by real-time deformability cytometry (RT-DC), and associated transcriptomics profiles are collected. (**B**) Target prediction. A subset of collected datasets is used to perform machine learning-based network analysis on transcriptomic data and identify conserved module of genes associated with cell mechanics changes. PC – principal component. (**C**) In silico validation. The classification performance of individual genes from module identified in (**B**) is evaluated in silico on remaining datasets. TPR – true positive rate, FPR – false positive rate, AUC – area under the curve. (**D**) Experimental validation. Targets with highest classification performance in silico are verified experimentally in perturbation experiments.

The online version of this article includes the following figure supplement(s) for figure 1:

**Figure supplement 1.** Characterization of mechanical cell properties using real-time deformability cytometry (RT-DC).

of the candidate genes, caveolin-1 (*CAV1*), has the capacity to alter the mechanical phenotype in the predicted direction when downregulated or overexpressed. The systematic approach presented here, combining omics data with mechanical phenotypes across different systems, has the power to identify genes that ubiquitously contribute to cell mechanical phenotype in a hypothesis-free manner. Such genes can, in the future, be used as knobs for adjusting mechanical cell properties to explore their role in the homeostasis of multicellular systems or to therapeutically intervene in relevant pathologies.

## Results

### Cross-system identification of genes involved in cell mechanical changes

We introduce an inference approach for de novo identification of genes involved in cell mechanical changes across different systems that we refer to as mechanomics. The general workflow of this approach is presented in *Figure 1* and consists of four steps: data curation, target prediction, in silico validation, and experimental validation. In the first step, mechano-transcriptomic datasets representing a broad spectrum of biological systems are collected (*Figure 1A*). Each dataset encompasses two or more cell states characterized by a distinct mechanical phenotype, for which transcriptomic data is available. In the second step, a subset of the transcriptomic datasets is used to identify a conserved network module of putative target genes involved in the regulation of cell mechanical phenotype (*Figure 1B*). The ability of the obtained target genes to correctly classify soft and stiff cell states is next tested in silico on the validation datasets (*Figure 1C*). Finally, the best scoring targets

**Table 1.** Mechano-transcriptomic datasets used in this study.

Pred – prediction, Val – validation, PI/II – positive hypothesis I/II, N – negative hypothesis, CCLE – cancer cell line encyclopedia, HT Seq – high-throughput RNA sequencing, CAGE – cap analysis of gene expression, AFM – atomic force microscopy, adeno – adenocarcinoma, wt – wild type, PP – proliferating progenitors, NNs – newborn neurons.

| General information | | | | Transcriptomic data | | | | | Mechanics data | |
| --- | --- | --- | --- | --- | --- | --- | --- | --- | --- | --- |
| Source | Dataset name | Used for | Cell states | Accession number | Reference | Method | Unique entries | Total samples used | Method | Reference |
| Human | Glioblastoma | Pred | FGFJI, EGF, serum | GEO: GSE77751 | *Poser et al., 2019* | HT seq | 39,400 | 27 | RT-DC | *Poser et al., 2019* |
| | Carcinoma | Val: PI | small-cell, adeno | DDBJ: DRA000991* | FANTOM5 *Forrest et al., 2014* | CAGE | 18,821 | 12 | RT-DC, AFM | this paper |
| | | Val: PII & N | | GEO: GSE36139† | CCLE microarray *Barretina et al., 2012* | Microarray | 18,925 | 162 | | |
| | | Val: PII | | DepMap: release 21Q4‡ | CCLE RNA-Seq *Ghandi et al., 2019* | HT seq | 51,304 | 179 | | |
| | | Val: PII & N | | GEO: GSE30611 | Genentech *Klijn et al., 2015* | HT seq | 25,996 | 82 | | |
| | MCF10A | Val: PI | wt, H1047R | GEO: GSE69822 | *Kiselev et al., 2015* | HT seq | 38,508 | 6 | RT-DC | this paper |
| Mouse | iPSCs | Pred | F-class, C-class | GEO: GSE49940 | *Tonge et al., 2014* | Microarray | 18,118 | 28 | RT-DC, AFM | *Urbanska et al., 2017* |
| | Developing neurons | Val: PI | PPs, NNs | GEO: GSE51606 | *Aprea et al., 2013* | HT seq | 21,110 | 9 | RT-DC | this paper |

*Data for samples of interest was extracted using TET tool from the FANTOM5 website https://fantom.gsc.riken.jp/5/.

†Data was downloaded using the ArrayExpress archive https://www.ebi.ac.uk/arrayexpress/experiments/E-MTAB-2706/.

‡DepMap Public 21Q4 Primary Files, accessed via DepMap portal https://depmap.org/portal/download.

are validated experimentally by monitoring mechanical phenotype changes upon their overexpression and downregulation in the cells of choice (*Figure 1D*).

## Model systems characterized by mechanical phenotype changes

To curate the mechano-transcriptomic datasets, we screened the projects ongoing in our group and identified five biological systems for which published transcriptomic data were available, and the concomitant mechanical phenotype changes were either already documented or implicated (*Table 1*). The mechanical phenotypes of the different cell states within each dataset were characterized primarily using real-time deformability cytometry (RT-DC), a microfluidics-based method that enables rapid analysis of thousands of cells (*Otto et al., 2015*; *Figure 1—figure supplement 1*) — a feature particularly useful when setting out to explore a large variety of systems and states. RT-DC relies on flowing cells through a narrow constriction of a microfluidic channel and high-speed imaging to assess the ensuing cell deformation (*Otto et al., 2015*; *Figure 1—figure supplement 1A, B*). In the context of this method, the mechanical phenotype is understood as whole-cell elasticity quantified by an apparent Young's modulus, $E$, deduced from cell size and deformation under given experimental conditions (*Mokbel et al., 2017*; *Figure 1—figure supplement 1C, D*). Young's modulus quantifies how much stress (force per unit area) is necessary to deform a cell to a certain extent (i.e., strain), thus higher Young's modulus values indicate that a cell is more difficult to deform, or stiffer. In two of the datasets (see *Table 1*), selected cell states were additionally characterized using atomic force microscopy (AFM)-based assays on adherent cells to confirm the mechanical differences observed with RT-DC. The transcriptional profiles related to each system, generated by either RNA sequencing (RNA-Seq) or microarray analysis, were retrieved from entries previously deposited in online databases (*Table 1*).

We curated mechano-transcriptomic data assemblies originating from five different biological systems (*Figure 2*) that included a total of eight transcriptomic datasets (*Table 1*). Two of the transcriptomic datasets were used for target prediction, and the reaming six for target validation. The first studied system encompassed patient-derived glioblastoma cell lines cultured in conditions supporting different levels of activation of the STAT3-Ser/Hes3 signaling axis involved in cancer growth regulation. As previously demonstrated, the higher the STAT3-Ser/Hes3 activation in the characterized states, the stiffer the measured phenotype of glioblastoma cells (*Poser et al., 2019*; *Figure 2A*). The second system included small-cell and adenocarcinoma cell lines originating from human intestine, lung, and stomach. Consistently across tissues, small cell-carcinoma cells had a lower apparent Young's modulus compared to their adenocarcinoma counterparts (*Figure 2B*). Small-cell carcinomas have comparatively small cell sizes, short doubling times and high metastatic potential, all connected with poor clinical prognosis in patients (*Brenner et al., 2004*; *Kalemkerian et al., 2013*). Apart from the main transcriptomic dataset for the carcinoma project, in which all mechanically characterized cell lines are represented (FANTOM5; *Forrest et al., 2014*), we collected three additional transcriptomic datasets generated with different expression profiling techniques (RNA-Seq or microarray profiling), and originating from different groups: Cancer Cell Line Encyclopedia (CCLE) microarray (*Barretina et al., 2012*), CCLE RNA-Seq (*Ghandi et al., 2019*), and Genentech (*Klijn et al., 2015*) (see *Table 1* for overview). In the third studied system, a non-tumorigenic breast epithelium MCF10A cell line bearing single-allele oncogenic mutation H1047R in the catalytic subunit alpha of the phosphatidylinositol-4,5-bisphosphate 3-kinase (PIK3CA) (*Juvin et al., 2013*; *Kiselev et al., 2015*) showed increased stiffness compared to the wild-type (WT) control (*Figure 2C*). H1047R mutation causes constitutive activation of PIK3CA and an aberrant triggering of the PI3K–AKT–mTOR signaling pathway leading to growth factor-independent proliferation (*Bader et al., 2006*; *Kang et al., 2005*). In the fourth system, the fuzzy-colony forming (F-class) state of induced pluripotent stem cells (iPSCs) had a lower stiffness as compared to the bone-fide compact-colony forming (C-class) state (*Urbanska et al., 2017*; *Figure 2D*). C-class cells establish endogenous expression of reprogramming factors at moderate levels toward the end of reprogramming, while F-class cells depend on the ectopic expression of the pluripotency factors and are characterized by a fast proliferation rate (*Tonge et al., 2014*). Finally, we characterized two stages of developing neurons isolated from embryonic mouse brain (*Aprea et al., 2013*), and observed that the newborn neurons (NNs) had higher apparent Young's moduli than proliferating progenitors (PPs) (*Figure 2E*). Cell areas and deformations used for Young's modulus extraction for all datasets are visualized in *Figure 2—figure supplement 1*.

The mechano-transcriptomic datasets collected within the framework of our study (*Table 1*) represent a broad spectrum of biological systems encompassing distinct cell states associated with mechanical phenotype changes. The included systems come from two different species (human and mouse), several tissues (brain, intestine, lung, stomach, breast, as well as embryonic tissue) and are associated with processes ranging from cancerogenic transformations to cell morphogenesis. This high diversity is important for focusing the analysis on genes universally connected to the change in mechanical properties, rather than on genes specific for processes captured by individual datasets.

## Discriminative network analysis on prediction datasets

After characterizing the mechanical phenotype of the cell states, we set out to use the accompanying transcriptomic data to elucidate genes associated with the mechanical phenotype changes across the different model systems. To this end, we utilized a method for inferring phenotype-associated functional network modules from omics datasets termed PC-corr (*Ciucci et al., 2017*), that relies on combining loadings obtained from the principal component (PC) analysis and Pearson's correlation for every pair of genes. PC-corr was performed individually on two prediction datasets, and the obtained results were overlayed to derive a conserved network module. Owing to the combination of the Pearson's correlation coefficient and the discriminative information included in the PC loadings, the PC-corr analysis does not only consider gene co-expression — as is the case for classical co-expression network analysis — but also incorporates the relative relevance of each feature for discriminating between two or more conditions; in our case, the conditions representing soft and stiff phenotypes. The overlaying of the results from two different datasets allows for a multiview analysis (utilizing multiple sets of features) and effectively merges the information from two different biological systems.

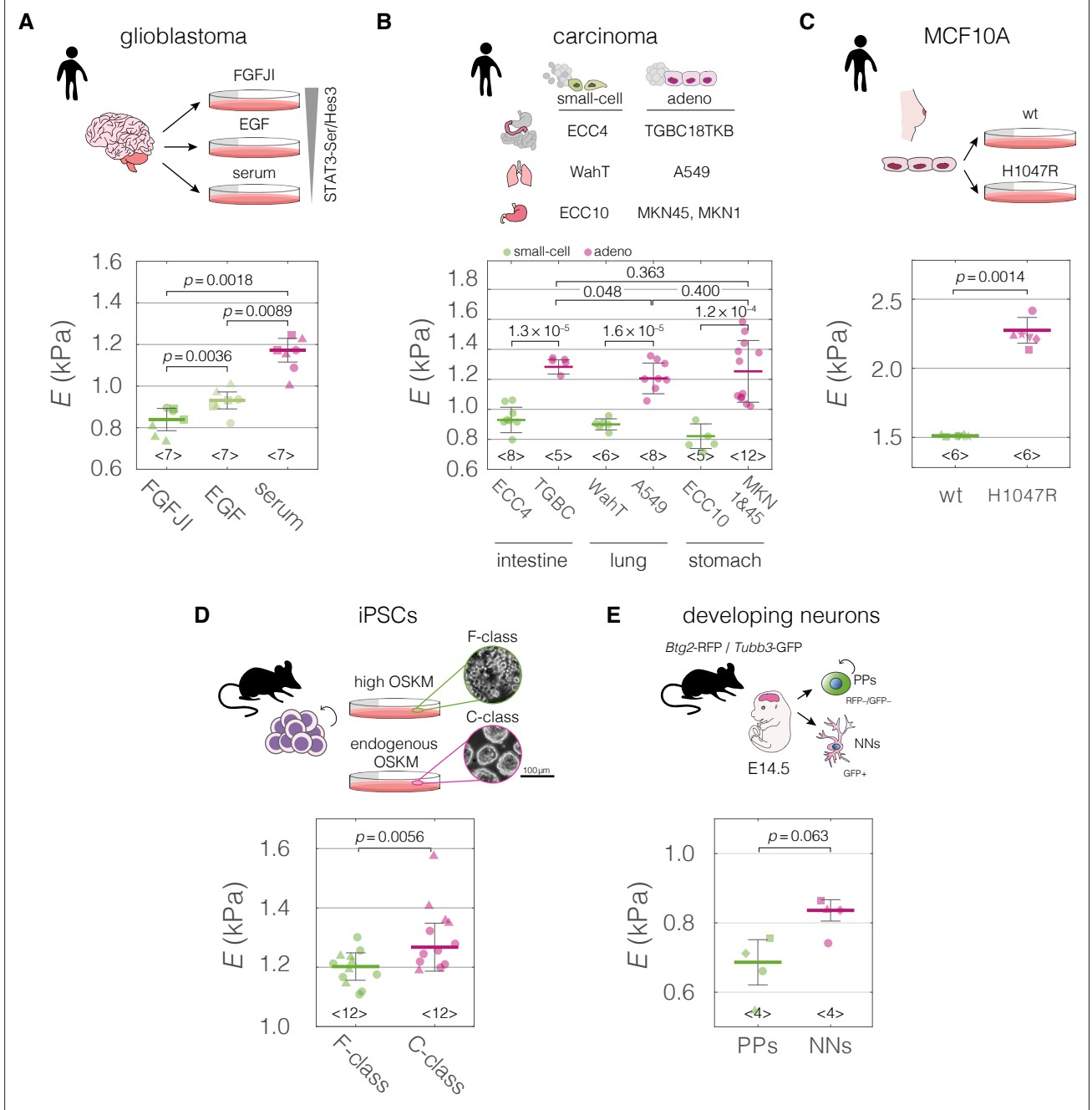

**Figure 2.** Mechanical properties of divergent cell states in five biological systems. Schematic overviews of the systems used in our study, alongside with the cell stiffness of individual cell states parametrized by Young's moduli $E$. (**A**) Human patient-derived glioblastoma cells with three distinct signaling states maintained by indicated culture conditions. (**B**) Human small-cell carcinoma and adenocarcinoma cell lines originating from intestine, lung, and stomach. (**C**) Human breast epithelium MCF10A cell line bearing single-allele H1047R mutation in the PIK3CA with parental wild type (wt) as a control. (**D**) Murine F- and C-class induced pluripotent stem cells (iPSCs) cultured in the presence (F-class) or absence (C-class) of doxycycline (dox) activating ectopic expression of OSKM factors (Oct4, Sox2, Klf4, and cMyc). (**E**) Proliferating progenitors (PPs) and newborn neurons (NNs) isolated from brains of mouse embryos. Horizontal lines delineate medians with mean absolute deviation (MAD) as error, datapoints represent medians of the individual measurement replicates, the number of independent biological replicates is indicated below each box. Statistical analysis was performed using generalized linear mixed effects model. The symbol shapes represent cell lines derived from three different patients (**A**), matched experimental replicates (**C**), two different reprogramming series (**D**), and four different cell isolations (**E**). Data presented in (**A**) and (**D**) were previously published in *Poser et al., 2019* and *Urbanska et al., 2017*, respectively. Cell areas and deformations used for Young's modulus extraction for all datasets are visualized in *Figure 2—figure supplement 1*.

The online version of this article includes the following source data and figure supplement(s) for figure 2:

*Figure 2 continued on next page*

*Figure 2 continued*

**Source data 1.** Young's moduli *E* for the datasets presented in *Figure 2A–E*.

**Figure supplement 1.** Plots of area vs deformation for different cell states in the characterized systems.

For the network construction, we chose two datasets that originate from different species, concern unrelated biological processes, and have a high number of samples included in the transcriptional analysis: human glioblastoma and murine iPSCs (*Table 1*). PC-corr analysis was performed on these prediction datasets individually using a subset of transcripts at which the two datasets intersect (*Figure 3A*). First, the 9452 unique genes from the intersection were used to perform principal component analysis (PCA) (*Figure 3B, C*). Next, the PC loadings for the component showing good separation between the different cell states (PC1 for both of presented datasets) were normalized and scaled (see Methods for details). The processed PC loadings, $V$, were then combined with Pearson's correlation coefficients, $c$, to obtain a *PC-corr* value for each pair of genes $i,j$ for every $n$-th dataset according to the following formula:

$$PC\text{-}corr_{i,j}^{n} = \text{sgn}(c_{i,j}^{n}) \min\left( \left|V_i^n\right|, \left|V_j^n\right|, \left|C_{i,j}^n\right| \right).$$ (1)

The sign of the *PC-corr* value corresponds to the correlated (positive) or anticorrelated (negative) expression of genes $i,j$, and the magnitude of *PC-corr* conveys the combined information about the strength of the expression correlation and the contribution of the individual genes to the phenotype-based separation of samples along the PC.

To merge the PC-corr results obtained for the individual prediction datasets (see *Figure 3D* for illustration), a combined *PC-corr* value, $PC\text{-}corr_{i,j}^{comb}$, was calculated either as a mean or as a minimum of the individual *PC-corr* values. For $n$ datasets:

$$PC\text{-}corr_{i,j}^{comb} = \begin{cases} \delta_{i,j} \dfrac{1}{N} \sum\limits_{n=1}^{N} \left| PC\text{-}corr_{i,j}^n \right| \\ \delta_{i,j} \min\left( \left| PC\text{-}corr_{i,j}^l \right|, \ldots, \left| PC\text{-}corr_{i,j}^n \right| \right) \end{cases}$$ (2)

where $\delta_{i,j} \in \{-1, 1\}$ defines the sign of $PC\text{-}corr_{i,j}^{comb}$, and is equal to the mode of $PC\text{-}corr_{i,j}$ signs over all individual datasets. In our implementation on two datasets, gene pairs with opposing *PC-corr* signs were masked by setting their $PC\text{-}corr^{comb}$ values to zero.

To obtain the network of putative target genes, a cut-off was applied to the absolute value of $PC\text{-}corr^{comb}$. We explored several cut-off strategies in order to obtain a wide overview of the meaningful conserved network structures. By looking at $PC\text{-}corr^{comb}$ calculated as mean and setting the threshold for its absolute value to 0.75, we obtained a network of 29 nodes connected by 30 edges (*Figure 3E*). The edges describe the connection between the genes in the network and their thickness is defined by the $PC\text{-}corr^{comb}$ values. The node colors reflect the strength of the contribution of individual genes to the separation of the different classes as described by the mean of the processed PC loadings .

The obtained network can be made more conservative by using the minimum $PC\text{-}corr^{comb}$ instead of the mean, or by changing the cut-off value. Utilizing the $PC\text{-}corr^{comb}$ calculated as minimum value and setting the cut-off value to 0.70, we obtained a network with 22 nodes connected by 29 edges (*Figure 3F*). Increasing the cut-off value to 0.75 resulted in a network of 9 genes connected by 12 edges (*Figure 3G*). The list of genes from the three networks presented in *Figure 3E–G*, together with their full names and processed PC loading values, is presented in *Figure 3—source data 2*.

We performed gene ontology enrichment analysis for biological processes on the nodes of the network presented in *Figure 3G*, as well as the union of all nodes presented in *Figure 3E–G* (*Figure 3—figure supplement 1*). The top two significantly enriched terms in the 9-gene set were the negative regulation of transcription by polymerase II (GO:000122) and negative regulation of endothelial cell proliferation (GO:0001937). In the 34-gene set, apart from a broad term of signal transduction (GO:0007165), the significantly enriched terms included negative regulation of transcription by polymerase II (GO:000122), regulation of cell growth (GO:0001558), and negative regulation of cell proliferation (GO:0008285), among others. The fact that these GO terms are not obviously related to cell mechanics might be an indicator that the association of the identified genes with cell mechanics

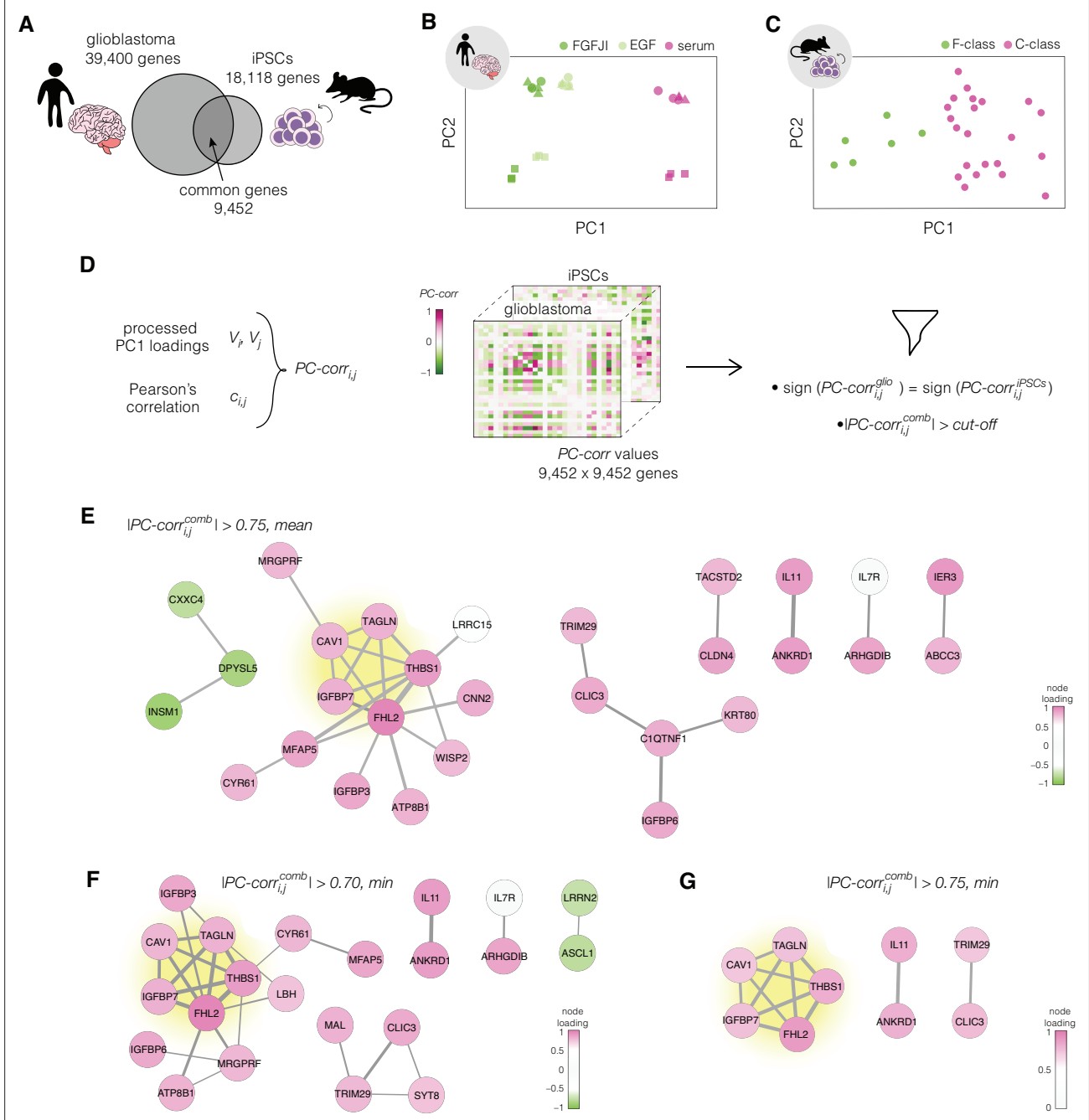

**Figure 3.** Identification of putative targets involved in cell mechanics regulation. (**A**) Glioblastoma and induced pluripotent stem cell (iPSC) transcriptomes used for the target prediction intersect at 9452 genes. (**B, C**) Principal component analysis (PCA) separation along two first principal components of the mechanically distinct cell states in the glioblastoma (**B**) and iPSC (**C**) datasets. The analysis was performed using the gene expression data from the intersection presented in (**A**). The symbol shapes in (**B**) represent cell lines derived from three different patients. (**D**) Schematic representation of PC-corr analysis and the combination of the PC-corr results for two systems. (**E–G**) Gene networks based on filtering gene pairs by the combined *PC-corr* score. The presented networks were obtained by setting the cut-off value to 0.75, when using the mean PC-corr approach (**E**), and to 0.70 (**F**) and 0.75 (**G**), when using the minimum value approach. In (**E–G**), edge thicknesses represent the $\left| PC\text{-}corr^{comb} \right|$ and the colors of the nodes represent the average processed PC loadings, both listed in *Figure 3—source data 2*.

The online version of this article includes the following source data and figure supplement(s) for figure 3:

**Source data 1.** PC1 and PC2 values for individual datapoints in *Figure 3B, C*.

**Source data 2.** Combined PC-corr values calculated as means or minimum value of the two discovery datasets, together with loadings of PC1, used for creating networks presented in *Figure 3E–G*.

**Figure supplement 1.** Gene ontology (GO) enrichment analysis of obtained target genes.

**Table 2.** List of identified target genes comprising the conserved module.

| Symbol | Gene description | HGNC ID | MGI ID |
| --- | --- | --- | --- |
| CAV1 | Caveolin-1 | HGNC:1527 | MGI:102709 |
| FHL2 | Four and a half LIM domains 2 | HGNC:3703 | MGI:1338762 |
| IGFBP7 | Insulin-like growth factor-binding protein 7 | HGNC:5476 | MGI:1352480 |
| TAGLN | Transgelin | HGNC:11553 | MGI:106012 |
| THBS1 | Thrombospondin 1 | HGNC:11785 | MGI:98737 |

is relative unknown, and that our mechanomics approach can identify such associations de novo. The aforementioned categories included mostly genes showing higher expression in the stiff states. Since the upregulated genes are associated with negative regulation of growth and transcription, our results point toward a targeted reduction in transcriptional activity and reduced growth/proliferation in stiff compared to soft cells.

## The identified conserved functional network module comprises five genes

Regardless of the strategy chosen for the selection of the network-building gene pairs, a strongly interconnected module of five genes (*Table 2*) — highlighted in yellow in *Figure 3E–G* — emerged. We focused on the five genes from this conserved network module as putative targets for regulating cell mechanics: *CAV1*, *FHL2*, *IGFBP7*, *TAGLN*, and *THBS1*.

Caveolin-1, CAV1, is a protein most prominently known for its role as a structural component of caveolae. Caveolae are small cup-shaped invaginations in the cell membrane that are involved, among other functions, in the mechanoprotective mechanism of buffering the plasma membrane tension (*Sinha et al., 2011*; *Parton and del Pozo, 2013*). Recent data suggests that CAV1 can also confer its mechanoprotective role independently of caveolae (*Lolo et al., 2023*). Apart from membrane organization and membrane domain scaffolding, CAV1 plays a role in an array of regulatory functions such as metabolic regulation or Rho-signaling (*Parton and del Pozo, 2013*). The second identified target, four and a half LIM domains 2, FHL2, is a multifaceted LIM domain protein with many binding partners and a transcription factor activity (*Johannessen et al., 2006*). FHL2 has recently been shown to remain bound to actin filaments under high tension, and be shuttled to the nucleus under low cytoskeletal tension (*Nakazawa et al., 2016*; *Sun et al., 2020*) — a property conserved among many LIM domain-containing proteins (*Sun et al., 2020*; *Winkelman et al., 2020*). The third target, Insulin-like growth factor-binding protein 7, IGFBP7, is a secreted protein implicated in a variety of cancers. It is involved in the regulation of processes such as cell proliferation, adhesion, and senescence (*Jin et al., 2020*). Transgelin, TGLN, is an actin-binding protein whose expression is upregulated by high cytoskeletal tension (*Liu et al., 2017*) and is also known to play a role in cancer (*Dvorakova et al., 2014*). Finally, thrombospondin 1, THBS1, is a matricellular, calcium-binding glycoprotein that mediates cell–cell and cell–matrix adhesions and has many regulatory functions (*Adams and Lawler, 2011*; *Huang et al., 2017*).

Before validating the performance of the five target genes, we inspected their expression across the divergent cell states in the collected datasets. The target genes show clear differences in expression levels between the soft and stiff cell states and provide for clustering of the samples corresponding to different cell stiffnesses in both prediction and validation datasets (*Figure 4*, *Figure 4—figure supplements 1–3*). The relation between normalized apparent Young's modulus change and fold-change in the expression of the target genes is presented in *Figure 4—figure supplement 4*. The direction of changes in the expression levels between the soft and stiff cell states in the validation datasets was not always following the same direction (*Figure 4C–F*, *Figure 4—figure supplement 4*). This suggests that the genes associated with cell mechanics may not have a monotonic relationship with cell stiffness, but rather are characterized by different expression regimes in which the expression change in opposite directions can have the same effect on cell stiffness. Additionally, in specific cases a relatively high change in Young's modulus did not correspond to marked expression changes of a given gene — see for example low CAV1 changes observed in MCF10A PIK3CA mutant (*Figure 4—figure

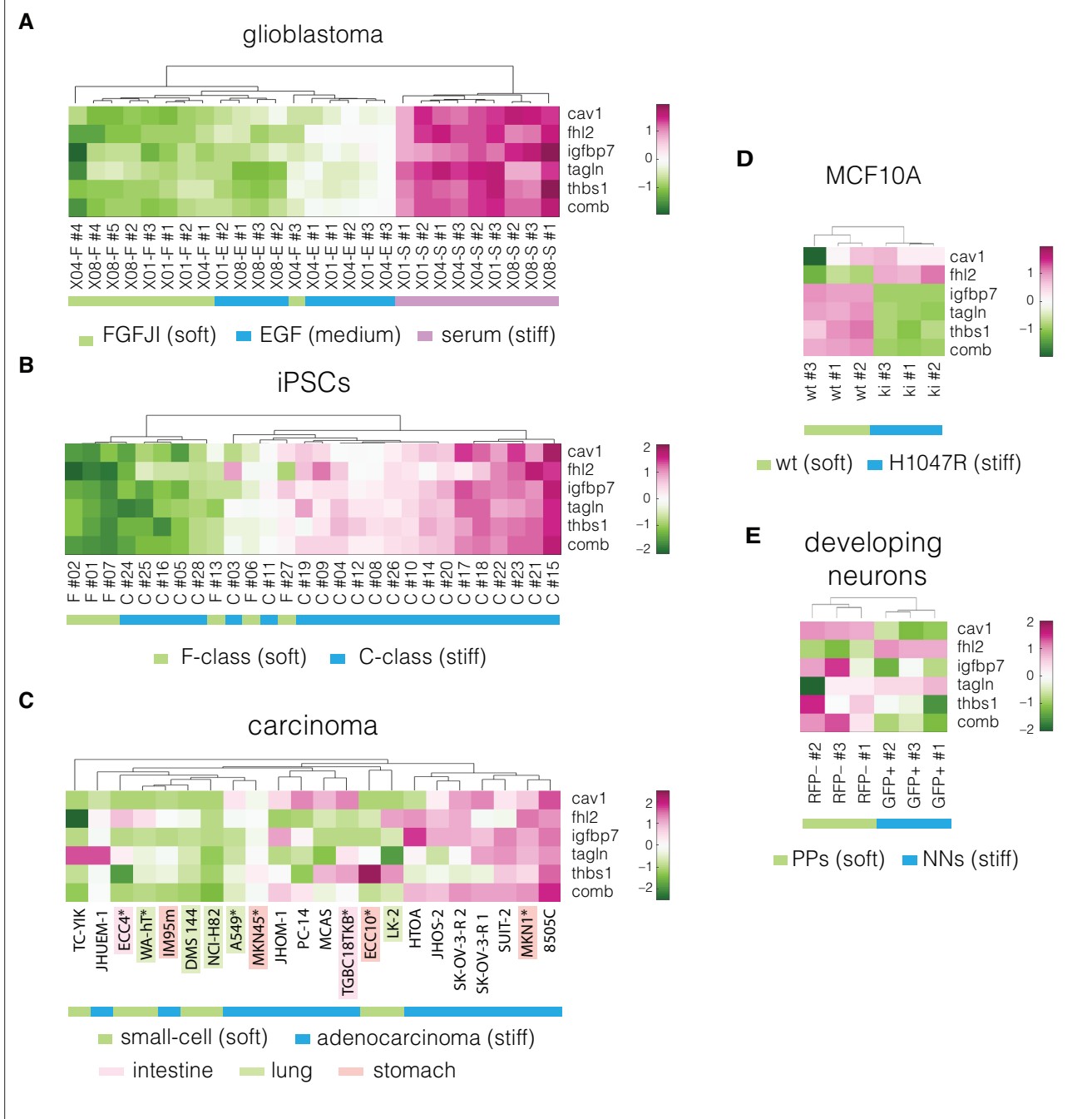

**Figure 4.** Expression of identified target genes in the prediction and validation datasets. Panels show unsupervised clustering heatmaps of expression data from transcriptomic datasets corresponding to the following systems: (**A**) glioblastoma, (**B**) induced pluripotent stem cells (iPSCs), (**C**) carcinoma, cell lines from intestine, lung, and stomach used for positive hypothesis I (see *Table 3*) are highlighted in pink, green, and orange, respectively; *mechanically tested cell lines (here the FANTOM5 dataset is presented as it contains all the cell lines that were tested mechanically in our study, for the remaining carcinoma datasets see *Figure 4—figure supplements 1–3*), (**D**) non-tumorigenic breast epithelia (MCF10A), and (**E**) developing neurons (dev. neurons). Comb – combinatorial marker, wt – wild type, PPs – proliferating progenitors, NNs – newborn neurons. Clustering was performed using *clustergram* function in *MATLAB* (*R2020a*,, MathWorks) on log-normalized expression data (*Figure 4—source data 1*).

The online version of this article includes the following source data and figure supplement(s) for figure 4:

**Source data 1.** Expression values of the target genes used for plotting the heatmaps in *Figure 4A–E*.

**Figure supplement 1.** Expression of identified target genes in the CCLE microarray dataset used for validation.

**Figure supplement 2.** Expression of identified target genes in the CCLE RNA sequencing (RNA-Seq) dataset used for validation.

*Figure 4 continued on next page*

*Figure 4 continued*

**Figure supplement 3.** Expression of identified target genes in the Genentech dataset used for validation.

**Figure supplement 4.** Relation between the magnitude of apparent Young's modulus change and the absolute change in the expression levels of target genes.

**Figure supplement 5.** Receiver-operator characteristics (ROC) curves characterizing classification performance of the five genes from the conserved module.

supplement 4A), or low IGFBP7 changes in intestine and lung carcinoma samples (*Figure 4—figure supplement 4C*). This indicates that the importance of specific targets for the mechanical phenotype change may vary depending on the origin of the sample.

## Universality, specificity, and trustworthiness of the identified markers

Next, we validated whether the five identified genes individually, as well as their association into a unique combinatorial marker (computed as the mean of the five log-normalized genes, see Methods), are universal and specific markers of cell mechanics. To assess that, we tested three hypotheses using combinations of transcriptomic data from six validation datasets as detailed in *Table 3*. The classification performance of each marker was assessed using the area under the curve of the receiver-operator characteristics (AUC-ROC) (*Hanley and McNeil, 1982*), which takes values from 0 to 1, with 1 corresponding to a perfect classifier and 0.5 to a random classifier. Importantly, for each hypothesis multiple datasets were used, and the discriminative performance was assessed in a joint multiview way by looking at the minimum value of AUC-ROC across multiple comparisons.

We first tested whether the obtained markers are universal across systems of different biological origin (positive hypothesis I) by estimating their ability to discriminate between stiff and soft cell phenotypes in three validation datasets: developing neurons (mouse), carcinoma cell lines originating from three tissues (human), and MCF10A (human) (*Table 4*). Particularly high minimum AUC-ROC values (≥0.78) were obtained for CAV1, FHL2, and TAGLN, and the combinatorial marker outperformed the individual genes with a minimum AUC-ROC of 0.97. The ROC curves for individual datasets are presented in *Figure 4—figure supplement 5*.

Next, we tested whether the identified markers provide good sample classification across similar datasets obtained from different sources (positive hypothesis II). For this purpose, we used three carcinoma datasets that were generated by two different research group using either microarray or RNA-Seq (see *Tables 1 and 3*). Within these datasets, we looked at the discrimination between the small-cell and adenocarcinoma samples from lung. This choice was dictated by the highest number of available samples from this tissue across the datasets. Also here, the multiview AUC-ROC values were high, reaching 0.89 for CAV1, 0.88 for FHL2, and 0.86 for THBS1. The combinatorial marker had an AUC-ROC value of 0.92.

To assess whether the predicted markers are specific to the mechanical phenotype, we tested their performance in classification of the adenocarcinoma samples grouped by the tissue they were derived from (negative hypothesis). These groups did not show clear mechanical differences (*Figure 2B*). For

**Table 3.** Overview of the hypotheses and datasets used for validating universality and specificity of obtained markers.

Hypotheses are listed in the column headings. Under every hypothesis, sample groups used for the hypothesis testing are listed. Numbers of samples used in every group are indicated in brackets.

| Positive hypothesis I: markers are discriminative of samples with stiff/ soft mechanical phenotype independent of the studied biological system | Positive hypothesis II: markers are discriminative of samples with stiff/soft mechanical phenotype independent of data source | Negative hypothesis: markers are discriminative of samples from different tissue of origin (but with no mechanical difference) |
|---|---|---|
| Carcinoma - FANTOM5 1. small-cell (*n* = 6) vs adeno (*n* = 6) (lung, intestine, and stomach) MCF10A 2. wt (*n* = 3) vs H1047R (*n* = 3) Developing neurons 3. PPs (*n* = 3) vs NNs (*n* = 3) | Carcinoma - CCLE microarray 1. small-cell (*n* = 51) vs adeno (*n* = 49) (lung) Carcinoma - CCLE RNA-Seq 2. small-cell (*n* = 51) vs adeno (*n* = 77) (lung) Carcinoma - Genentech (RNA-Seq) 3. small-cell (*n* = 30) vs adeno (*n* = 38) (lung) | Carcinoma - CCLE microarray 1. lung (*n* = 49) vs stomach (*n* = 19)(adeno) 2. large intestine (*n* = 43) vs stomach (*n* = 19) (adeno) Carcinoma - Genentech (RNA-Seq) 3. lung (*n* = 38) vs stomach (*n* = 14) (adeno) |

**Table 4.** Validation of identified target genes and the combinatorial marker.

Minimum AUC-ROC (min AUC-ROC) and JVT p values are reporter for the two positive hypotheses and one negative hypothesis for each target genes and the combinatorial marker (comb, highlighted in bold). The specific datasets and comparisons used for testing of each hypothesis are listed in *Table 3*. The results presented in this table can be reproduced using the code and data available on GitHub as reported in the Materials and methods.

|  | Measure | *CAV1* | *FHL2* | *IGFBP7* | *TAGLN* | *THBS1* | comb |
|---|---|---|---|---|---|---|---|
|  | min AUC-ROC | 0.78 | 0.89 | 0.67 | 0.78 | 0.56 | **0.97** |
| Positive hypothesis I | JVT p value | 0.14 | 0.04 | 0.30 | 0.14 | 0.81 | **0.01** |
|  | min AUC-ROC | 0.89 | 0.88 | 0.73 | 0.56 | 0.86 | **0.92** |
| Positive hypothesis II | JVT p value | 0.02 | 0.03 | 0.19 | 0.59 | 0.04 | **0.01** |
|  | min AUC-ROC | 0.54 | 0.51 | 0.51 | 0.52 | 0.61 | **0.51** |
| Negative hypothesis | JVT p value | 0.40 | 0.76 | 0.90 | 0.61 | 0.06 | **0.91** |

the combinatorial marker, the min AUC-ROC value was equivalent to a random classifier (0.51), and for the individual markers reached values between 0.51 and 0.65 (*Table 4*). Since the discriminative power of the obtained markers vanished (reached AUC-ROC close to 0.50 corresponding to a random classifier) when tested on groups that do not encompass cell mechanic phenotype difference, we can conclude that the identified markers are specific to the mechanical phenotype.

Finally, to test the trustworthiness of obtained markers, we evaluated how easy it is to generate markers with equivalent discriminative power at random. For that purpose, we devised a novel methodology called joint-view trustworthiness (JVT). JVT is a resampling technique that creates a null model distribution according to which an empirical p value is computed to evaluate the probability to sample at random a marker that offers a joint multview discrimination equal or better to the one of the predicted markers (see Methods for details). A low JVT p value (<0.05 significance level) means that it is rare to randomly generate a joint multiview marker with performance equal or better than the tested one. As summarized in *Table 4*, the combinatorial marker had remarkably low JVT p values (p = 0.01) in positive hypotheses I and II, that is, it is very unlikely to generate a similarly performing combinatorial marker at random. Conversely, in the negative hypothesis, the JVT p value of the combinatorial marker is not significant (p = 0.91). The performance of the tested genes individually was varied, with FHL2 showing a significant JVT p value in positive hypothesis I, and FHL2, CAV1, and THBS1 reaching significant JVT p values in positive hypothesis II. It is important to note that our implementation of JVT is conservative, as we consider the minimum discriminative performance on multiple datasets. This may lead to underestimating the performance of individual markers. In sum, the results provided in *Table 4* pointed toward CAV1 and FHL2 as promising markers of the mechanical phenotype.

## Perturbing expression levels of CAV1 changes cells stiffness

We decided to focus our attention on CAV1 as a potential target for modulating mechanical properties of cells, as it has previously been linked to processes intertwined with cell mechanics. In the context of mechanosensing, CAV1 is known to facilitate buffering of the membrane tension (*Sinha et al., 2011*), play a role in β1-inegrin-dependent mechanotransduction (*del Pozo et al., 2005*) and modulate the mechanotransduction in response to substrate stiffness (*Moreno-Vicente et al., 2018*). CAV1 is also intimately linked with actin cytoskeleton — it was shown to be involved in cross-talk with Rho-signaling and actin cytoskeleton regulation (*Parton and del Pozo, 2013*; *Raudenska et al., 2020*; *Pol et al., 2020*; *Lin et al., 2015*), filamin A-mediated interactions with actin filaments (*Muriel et al., 2011*), and co-localization with peripheral actin (*Sun et al., 2003*). The evidence directly relating CAV1 levels with the mechanical properties of cells (*Lolo et al., 2023*; *Lin et al., 2015*; *Hsu et al., 2018*; *Le Master et al., 2022*) and tissues (*Le Master et al., 2022*; *Grivas et al., 2020*), is only beginning to emerge.

In most of the mechano-transcriptomic datasets considered in our study, the increase in apparent Young's modulus was accompanied by an increase in CAV1 levels (*Figure 4—figure supplement 4A*), corroborating previous reports (*Lin et al., 2015*; *Hsu et al., 2018*; *Le Master et al., 2022*). Additionally, we observed that mouse embryonic fibroblasts isolated from CAV1 knock out mice (CAV1KO) are softer than the WT cells (*Figure 5—figure supplement 1*). Thus, we set out to test weather artificially

decreasing the levels of CAV1 results in cell softening, and conversely, increasing the level of CAV1 in higher cell stiffness. To this end, we perturbed the levels of CAV1 in the cell lines representing two intestine carcinoma types: ECC4, the small-cell carcinoma with a comparably soft phenotype, and TGBC18TKB (TGBC), the adenocarcinoma with a comparatively stiff phenotype.

Before perturbations, we confirmed that TGBC cells have higher levels of CAV1 compared to ECC4 cells on a protein level (*Figure 5A*), and that they are characterized by a stiffer phenotype, not only when measured with RT-DC (*Figures 2B and 5B*), but also with AFM using both standard indentation experiments (*Figure 5C*), as well as oscillatory measurements at different frequencies, referred to as AFM microrheology (*Figure 5D*).

The three techniques for characterizing mechanical properties of cells — RT-DC, AFM indentation, and AFM microrheology — differ in several aspects (summarized in *Supplementary file 1*), most notably in the frequency at which the force is applied to cells during the measurements, with RT-DC operating at the highest frequency (~600 Hz), AFM microrheology at a range of frequencies in-between (3–200 Hz), and AFM indentation operating at lowest frequency (5 Hz) (see *Supplementary file 1* and *Figure 5—figure supplement 2A*). Even though the apparent Young's moduli obtained for TGBCS cells were consistently higher than those for ECC4 cells across all three methods, the absolute values measured for a given cell line varied depending on the methods: RT-DC measurements yielded higher apparent Young's moduli compared to AFM indentation, while the apparent Young's moduli derived from AFM microrheology measurements were frequency-dependent and fell between the other two methods (*Figure 5B–D*, *Figure 5—figure supplement 2B*). The observed increase in apparent Young's modulus with probing frequency aligns with previous findings on cell stiffening with increased probing rates observed for both AFM indentation (*Li et al., 2008*; *Zhou et al., 2012*) and microrheology assays (*Alcaraz et al., 2003*; *Massiera et al., 2007*; *Rigato et al., 2017*).

To decrease the levels of CAV1 in the TGBC cells, we performed knock-down experiments using two RNAi systems, endoribonuclease-prepared siRNA (esiRNA) targeting three different parts of CAV1 transcript (esiCAV1-1, esiCAV1-2, and esiCAV1-3), and a pool of conventional siRNAs (CAV1-pool) (*Figure 5E*). All the RNAi approaches resulted in the decrease of the apparent Young's modulus of TGBC cells as measured by RT-DC (*Figure 5F*, *Figure 5—figure supplement 3A, B*). The most prominent effect was observed using esiCAV1-1. We further confirmed that CAV1 knock-down with esiCAV1-1 resulted in decreased stiffness of TGBC cells using AFM indentation (*Figure 5G*) and microrheology (*Figure 5H*) (for overview of the results from all three methods see *Figure 5—figure supplement 2C*).

To investigate the influence of increased CAV1 levels on cell stiffness, we performed transient overexpression experiments of CAV1 with a dTomato reporter under independent ribosomal entry site, IRES, (CAV1iT) in both ECC4 and TGBC cell lines. At 72 hr post transfection, we observed elevated levels of CAV1 in both cell lines on a protein level in bulk (*Figure 5I*). Since in the transient overexpression experiments not all of the cells are transfected, we leveraged the possibility to monitor the fluorescence of single cells in parallel with their mechanical phenotype offered by real-time fluorescence and deformability cytometry (RT-FDC) (*Rosendahl et al., 2018*) to gate for the fluorescence-positive cells (T+, gate marked in magenta in *Figure 5J*). The fluorescence-positive cells in the CAV1-transfected sample, CAV1iT+, showed higher apparent Young's moduli as compared to fluorescence-negative cells in both control sample (mock) and CAV1-transfected sample (CAV1iT–, internal control) (*Figure 5J*, *Figure 5—figure supplement 3C, D*). The effect was observed in ECC4 as well as TGBC cells. However, it was more pronounced in the TGBC cells, suggesting that the cells may be more responsive to the artificial increase in CAV1 levels when natively expressing a basal level of this protein.

Finally, we performed CAV1 perturbation experiments in a breast epithelial cell model of cancerous transformation, MCF10A-ER-Src cells, in which the Src proto-oncogene can be induced by treatment with tamoxifen (TAM). As previously shown, TAM addition triggers Src phosphorylation and cellular transformation (*Hirsch et al., 2009*), which is associated with F-actin cytoskeletal changes and, after a transient stiffening, the acquisition of a soft phenotype evident at 36 hr post induction (*Tavares et al., 2017*). We inspected a previously published microarray dataset and determined that the expression of CAV1 diminishes over time after TAM treatment (*Hirsch et al., 2010*; *Figure 6A*). We then showed that the decrease of CAV1 could be observed at the protein level 72 hr post induction (*Figure 6B*), a timepoint at which the TAM-induced MCF10A-ER-Src cells show a significant decrease in cell stiffness

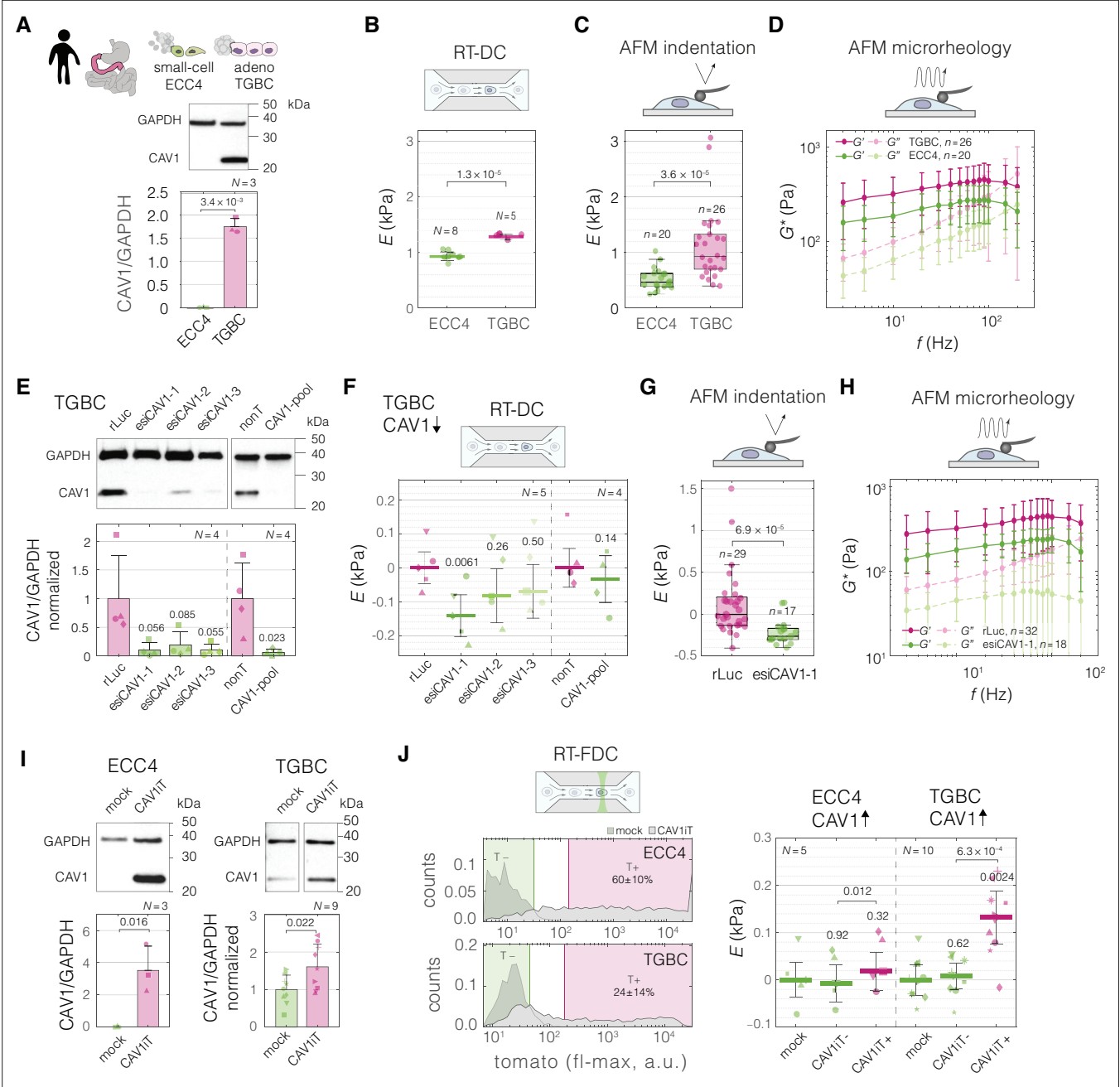

**Figure 5.** Perturbing levels of CAV1 affects the mechanical phenotype of intestine carcinoma cells. (**A**) CAV1 levels in small-cell (ECC4) and adenocarcinoma (TGBC) cell lines from intestine. Mechanical phenotype of ECC4 and TGBC cells measured with real-time deformability cytometry (RT-DC) (**B**, as in *Figure 2B*), atomic force microscopy (AFM) indentation (**C**), and AFM microrheology (**D**). (**E**) Protein-level verification of CAV1 knock-down in TGBC cells using two knock-down system: three esiRNA constructs (esiCAV1-1. esiCAV1-1, and esiCAV1-3 with rLuc as a control), and pooled siRNA mixture (CAV1-pool with non-targeting mixture nonT as a control). Mechanical phenotype change of TGBC cells upon CAV1 knock-down as measured by RT-DC (**F**), AFM indentation (**G**), and AFM microrheology (**H**). (**I**) Protein-level verification of transient CAV1 overexpression in ECC4 and TGBC cells. (**J**) Mechanical phenotype change of ECC4 and TGBC cells upon CAV1 overexpression as measured by real-time fluorescence and deformability cytometry (RT-FDC). Gating for fluorescence-positive and -negative cells based on dTomato expression in ECC4 (top) and TGBC (bottom) cells (left-hand side). Fluorescence-positive cells correspond to cells expressing CAV1-IRES-dTomato (CAV1iT). For comparison, mock transfection sample is shown in the background (mock). Apparent Young's modulus changes of ECC4 and TGBC cells upon CAV1 overexpression (right-hand side). CAV1iT− and CAV1T+ are dTomato negative and positive cells, respectively. For protein quantification in (**A, E, and I**), representative western blots (top) as well as quantification of specified replicate numbers N (bottom) are shown. In (**B, F, and J**), horizontal lines delineate medians with mean absolute deviation (MAD) as error, datapoints represent medians of N experiment replicates, statistical analysis was performed using generalized linear mixed effects model. In (**C**) and (**G**), box plots spread from 25th to 75th percentiles with a line at the median, whiskers span 1.5 × interquartile range (IQR), individual

*Figure 5 continued on next page*

*Figure 5 continued*

datapoints correspond to values obtained for *n* individual cells, statistical analysis was performed using two sample two-sided Wilcoxon rank sum test. In (**D**) and (**H**), datapoints correspond to means ± standard deviation of all measurements at given oscillation frequencies for *n* cells. Lines connecting datapoints serve as guides for the eye. *E* – apparent Young's modulus, *G\** – complex shear modulus, *ΔE* – apparent Young's modulus change relative to respective control measurements. In (**E, F, I, and J**), the symbol shapes represent matching experiment replicates.

The online version of this article includes the following source data and figure supplement(s) for figure 5:

**Source data 1.** CAV1 protein levels presented in *Figure 5A, E and I*.

**Source data 2.** Mechanical measurements conducted in the perturbation experiments on ECC4 and TGBC cell lines using real-time deformability cytometry (RT-DC), atomic force microscopy (AFM) indentation, and AFM oscillatory measurements.

**Source data 3.** FL2-max data for the histograms presented in *Figure 5J*.

**Source data 4.** Original membrane scans for all replicates.

**Source data 5.** Overview of all blots with labelled protein size markers and bands.

**Figure supplement 1.** CAV1 knock-out mouse embryonic fibroblasts (CAV1KO) have lower stiffness compared to the wild-type (WT) cells.

**Figure supplement 1—source data 1.** Original membrane scans for all replicates.

**Figure supplement 1—source data 2.** Overview of all blots with labelled protein size markers and bands.

**Figure supplement 2.** Absolute Young's modulus values across the probing frequencies characteristic for the three measurement methods.

**Figure supplement 3.** Plots of area vs deformation from real-time deformability cytometry (RT-DC) measurements of cells with perturbed CAV1 levels.

(*Tavares et al., 2017* and *Figure 6C*). We next showed that knocking down CAV1 decreased the stiffness of uninduced MCF10A-ER-Src cells (*Figure 6D*), similar to the effect of TAM induction. Finally, we performed an inverse experiment, in which we rescued the CAV1 levels in TAM-induced MCF10A-ER-Src cells by transient overexpression. The cells with CAV1 overexpression showed a stiff phenotype, corresponding to the one of uninduced cells (*Figure 6E*).

Taken together, the results obtained with the intestine carcinoma cell lines and MCF10A-ER-Src cells show that CAV1 not only correlates with, but also is causative of mechanical phenotype change.

## Discussion

The mechanical phenotype of cells is recognized as a hallmark of many physiological and pathological processes. Understanding how to control it is a necessary next step that will facilitate exploring the impact of cell mechanics perturbations on cell and tissue function (*Guck, 2019*). The increasing availability of transcriptional profiles accompanying cell state changes has recently been complemented by the ease of screening for mechanical phenotypes of cells thanks to the advent of high-throughput microfluidic methods (*Urbanska et al., 2020*). This provides an opportunity for data-driven identification of genes associated with the mechanical cell phenotype change in a hypothesis-free manner. Here, we leveraged this opportunity by performing discriminative network analysis on transcriptomes associated with mechanical phenotype changes to elucidate a conserved module of five genes potentially involved in cell mechanical phenotype regulation. We provided evidence that the inferred conserved functional network module contains an ensemble of five genes that, in particular when combined in a unique combinatorial marker, are universal, specific and trustworthy markers of mechanical phenotype across the studied mouse and human systems. We further demonstrated on the example of a selected marker gene, CAV1, that its experimental up- and downregulation impacts the stiffness of the measured cells. This demonstrates that the level of CAV1 not only correlates with, but also is causative of mechanical phenotype change. The mechanistic insights into how precisely the identified genes are involved in regulating mechanical properties, how they interact with each other, and whether they are universal and dominant in various contexts all remain to be established in future studies.

The workflow presented here is a blueprint for data-driven discovery of cell mechanics markers that can serves as targets for modulating cell mechanical properties. Its key features are the hypothesis-free modus operandi and the integration of information from different biological systems, that allows to focus on genes that play a relatively general role in cell mechanics rather than on genes specific to the individual experimental models. Noteworthy, by including the PC loadings in the scores used for thresholding, the PC-corr method implemented for network analysis in our study offers a multivariate alternative to classical co-expression analysis, that highlights not only the correlation between the

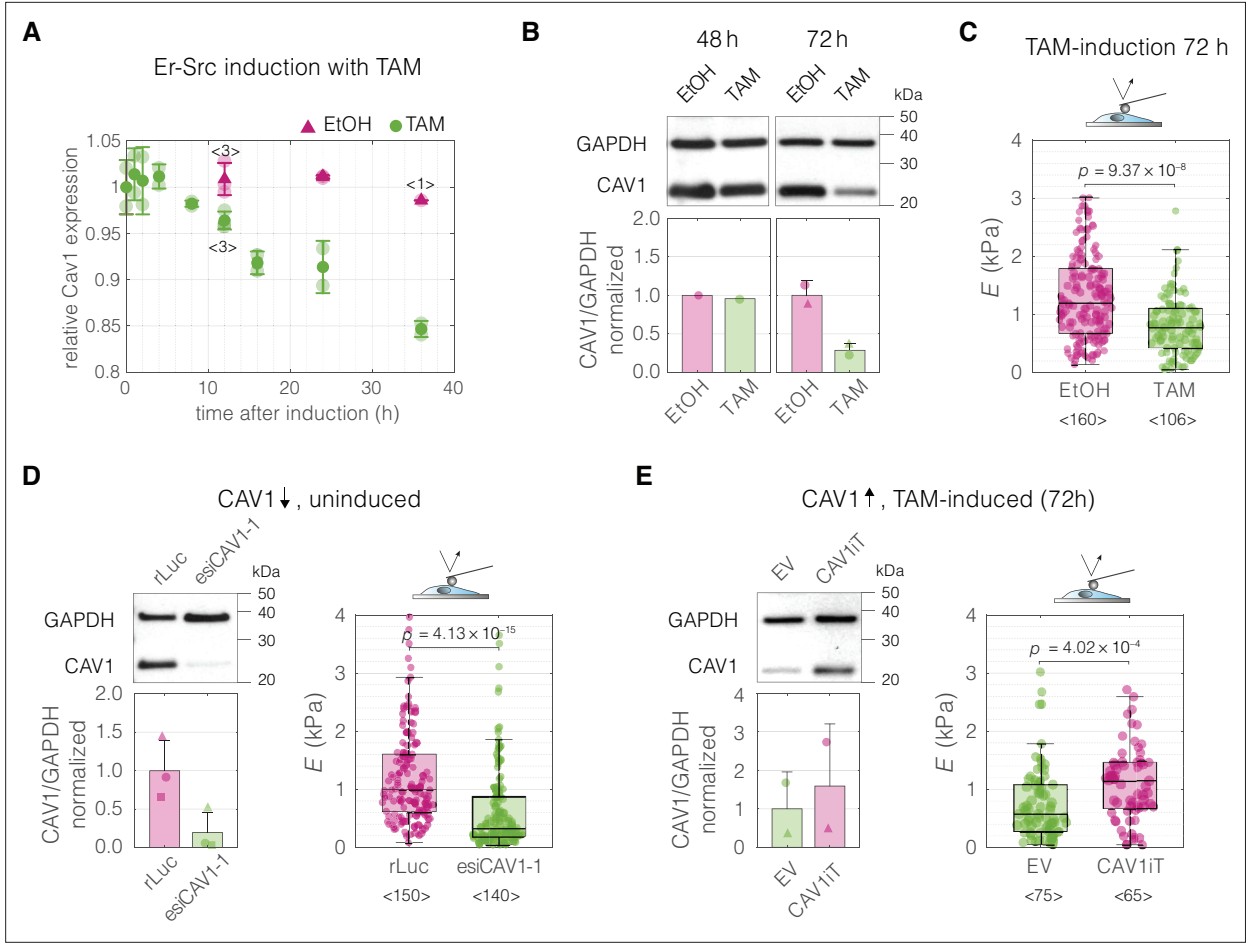

**Figure 6.** Perturbations of CAV1 levels in MCF10A-ER-Src cells result in cell stiffness changes. (**A**) Inducing transformation of MCF10A- ER-Src cells by tamoxifen (TAM) treatment, as opposed to vehicle control (ethanol, EtOH), causes a decrease of CAV1 expression over time, as captured by microarray analysis (GEO accession number: GSE17941, data previously published in *Hirsch et al., 2010*). Datapoints with error bars represent means ± standard deviation ($N = 2$, unless indicated otherwise). (**B**) Western blot analysis shows the decrease of CAV1 at protein level 72 hr post induction. (**C**) MCF10A-ER-Src cells show decreased apparent Young's moduli 72 hr post TAM induction. (**D**) CAV1 knock-down in uninduced MCF10A-ER-Src cells results in lowering of the apparent Young's modulus. (**E**) Overexpression of CAV1 in TAM-induced MCF10A-ER-Src cells causes increase in the apparent Young's modulus and effectively reverts the softening caused by TAM induction (compare to panel C). Box plots in (C–E) spread from 25th to 75th percentiles with a line at the median, whiskers span 1.5 × interquartile range (IQR), individual datapoints correspond to values obtained for individual cells, the number of measured cells per conditions, pooled from $N = 3$ independent experiments, is indicated below each box. Statistical analysis was performed using a two-sided Wilcoxon rank sum test. In the bar graphs in (**B, D, and E**), the symbol shapes represent experiment replicates.

The online version of this article includes the following source data for figure 6:

**Source data 1.** CAV1 expression and protein levels associated with MCF10A-Er-Src perturbation experiments presented in *Figure 6A, B, D, and E*.

**Source data 2.** Young's moduli $E$ obtained from atomic force microscopy (AFM) indentation measurements for the MCF10A-Er-Src perturbation experiments presented in *Figure 6C–E*.

**Source data 3.** Original membrane scans for all replicates.

**Source data 4.** Overview of all blots with labelled protein size markers and bands.

genes but also their relative importance for separating samples based on their mechanical phenotype. Despite its simplicity, PC-corr offers a robust performance on different types of omics data, and has already proven its efficacy in several studies (*Ciucci et al., 2017*; *Poser et al., 2019*; *Durán et al., 2021*).

The mechanical phenotype of single cells is a global readout of cell's resistance to deformation that integrates contributions from all cellular components. The three techniques implemented for measuring cell mechanics in this study — RT-DC, AFM indentation, and AFM microrheology — exert comparatively low deformations (<3 μm, see *Supplementary file 1*), at which the actin cortex is

believed to dominate the measured response. However, other cellular components, including the membrane, microtubules, intermediate filaments, nucleus, other organelles, and cytoplasmic packing, can also contribute to the measured deformations (reviewed in detail in *Urbanska and Guck, 2024*) and, for a particular system, it is hard to speculate without further investigation which parts of the cell have a dominant effect on the measured deformability.

Among the target genes elucidated in our analysis, we did not observe enrichment of gene ontology terms related to actin cytoskeleton organization, actomyosin contractility, or cell migration — processes that are typically associated with cell mechanics (*Figure 3—figure supplement 1*). This can be partially explained by looking at the mRNA rather than the protein level, its supramolecular assembly, activation state or localization. Upon closer inspection of the obtained gene targets, we found some links connecting them with cell mechanics in the literature. As indicated above, CAV1 has been shown to be involved in cross-talk with Rho-signaling and actin-related processes, as well as physical interactions with actin (*Parton and del Pozo, 2013*; *Raudenska et al., 2020*; *Pol et al., 2020*; *Lin et al., 2015*; *Muriel et al., 2011*; *Sun et al., 2003*). It is thus conceivable that CAV1 is involved in cell mechanics regulation via its influence on the actin cytoskeleton and its contractility. Furthermore, CAV1 is known to modulate the activation of transcriptional cofactor yes-associated protein, YAP, in response to changes in stiffness of cell substrate (*Moreno-Vicente et al., 2018*) and in the mechanical stretch-induced mesothelial to mesenchymal transition (*Strippoli et al., 2020*). YAP is an established transducer of not only various mechanical stimuli, but also of cell shape and the changes in the actin cytoskeleton tension (*Dupont et al., 2011*), the latter being an important determinant of cell stiffness. Conversely, YAP is an essential co-activator of CAV1 expression (*Rausch et al., 2019*). In the extended networks (*Figure 3E, F*, *Figure 3—source data 2*), we found three further genes that are identified (CYR61 and ANKRD1) (*Stein et al., 2015*; *Zhao et al., 2008*) or implicated (THBS1) (*Dupont et al., 2011*) as transcriptional targets of YAP. The next identified marker gene, transgelin, TGLN (also known as SM22α) is an actin-binding protein, that stabilizes actin filaments and is positively correlated with cytoskeletal tension (*Jiang et al., 2014*). Transgelin is a member of the calponin protein family, one further member of which, calponin 2, CNN2, is present in the broader sets of genes identified in this study (*Figure 3E, F*, *Figure 3—source data 2*). The expression of calponin 2, likewise, stabilizes actin filaments and is increased in cells with high cytoskeletal tension (*Hossain et al., 2005*). Finally, FHL2 is a transcriptional co-activator that is found, together with other LIM domain protein families such as zyxin and paxillin, to localize to actin filaments that are under stress (*Nakazawa et al., 2016*; *Sun et al., 2020*; *Winkelman et al., 2020*). When the cytoskeletal tension is low, FHL2 translocates to the nucleus, thus serving as a nuclear transducer of actomyosin contractility (*Nakazawa et al., 2016*).

To our knowledge, there are no prior studies that aim at identifying gene signatures associated with single-cell mechanical phenotype changes, in particular across different cell types. There are, however, several studies that investigated changes in expression upon exposure of specific cell types to mechanical stimuli such as compression (*De Marzio et al., 2021*; *Kılıç et al., 2020*) or mechanical stretch (*Zhang et al., 2021*; *Strippoli et al., 2020*; *Rysä et al., 2018*), and one study that investigated difference in expression profiles between stiffer and softer cells sorted from the same population (*Lv et al., 2021*). Even though the studies concerned with response to mechanical stimuli answer a fundamentally different question (how gene expression changes upon exposure to external forces vs which genes are expressed in cells of different mechanical phenotype), we did observe some similarities in the identified genes. For example, in the differentially expressed genes identified in the lung epithelia exposed to compression (*De Marzio et al., 2021*), three genes from our module overlapped with the immediate response (CAV1, FHL2, and TGLN) and four with the long-term one (CAV1, FHL2, TGLN, and THBS1). We speculate that this substantial overlap is caused by the cells undergoing change in their stiffness during the response to compression (and concomitant unjamming transition). Another previous study explored the association between the stiffness of various tissues and their proteomes. Despite the focus on the tissue-scale rather than single-cell elasticity, the authors identified polymerase I and transcript release factor (PTRF, also known as cavin 1 and encoding for a structural component of the caveolae) as one of the proteins that scaled with tissue stiffness across samples (*Swift et al., 2013*).

As seen from the example of the target genes included in the conserved module, their change is correlated with cell mechanics across all datasets, but it does not always follow the same trend (*Figure 4*, *Figure 4—figure supplement 4*). This non-monotonic relationship between gene

expression and the mechanical phenotype change suggests that there may be different regimes at which the expression change in the same direction has an opposite effect on the property of interest. Furthermore, the effect of expression change may be contextual and depend on the state of cells. This observation carries some parallels to the role of several of our target genes in cancer progression. For example, CAV1 has been indicated as both promoting and suppressing cancer progression in a variety of tissues. One way in which this can be reconciled is that the change in CAV1 expression may have different roles depending on the stage of caner progression (*Raudenska et al., 2020*; *Goetz et al., 2008*; *Wang et al., 2015*). A similar ambiguity of their role in cancer progression was indicated for THBS1 (*Huang et al., 2017*) and IGFBP7 (*Jin et al., 2020*). Of note, a non-monotonic cell stiffness response has also been described for treatments with actin-disrupting drugs. For example, treating cells with Latrunculin B makes cells progressively more deformable up to a certain concentration, beyond which the cells become less deformable again and eventually even stiffer than non-treated cells (see *Urbanska et al., 2020*) and discussion therein for more references. Apart from characterizing the response regimes, it will be also important to consider the temporal dynamics of cell response to the change in expression of a given gene. Trying to push the cell out of its equilibrium may cause the system to respond actively to counterbalance the induced change, which, in turn, may lead to oscillations in both expression levels of manipulated protein and its effectors, as well as the mechanical properties of the cell.

Among all different types of omics data, looking at the transcriptome is advantageous and disadvantageous at the same time. Its limitation is that mRNA levels do not necessarily reflect protein content in cells. Furthermore, for many proteins it is not the absolute level that has a functional relevance, but rather the protein activation by, for example, phosphorylation or binding with co-activators, or its localization. However, identifying the players at the transcriptome level has the advantage of easy implementation in perturbation experiments with established genetic tools, such as CRISPR–Cas9 technology or RNAi. Our analysis framework is readily applicable to other types of omics data, including proteomic, metabolomic, lipidomic, or glycomic data, the analysis of which would complement our study and provide different insights into the regulation of cell mechanics. Lipidomic data, for example, could reveal possible contributors to cell mechanics related to the composition of the cell membrane.

For the approaches such as the one pioneered in this study to flourish, it is necessary that the mechanical datasets become routinely published and annotated in a manner similar to omics datasets. With the recent advent of high-throughput cell mechanical characterization techniques, such as deformability cytometry methods (*Urbanska et al., 2020*), the establishment of a database for cell mechanics gains immediate relevance. In our group alone, within the timespan of 9 years since the RT-DC method was originally published (*Otto et al., 2015*), we have accumulated over 200,000 individual mechanical characterization experiments, comprising roughly two billion of single cells measured. Once a vast number of mechanics datasets connected to omics profiles is available, it will be straightforward to develop a next generation artificial intelligence algorithm predicting cell stiffness from given omics profiles. Apart from analyzing divergent cell states, the search for mechanical regulators could be complemented by looking into omics data of cells from unimodal populations sorted by their mechanical properties — a pursuit that with the advent of high-throughput methods for mechanics-based sorting of cells, such as sorting RT-DC (*Nawaz et al., 2020*; *Nawaz et al., 2023*) or passive filtration-based approaches (*Lv et al., 2021*), becomes a realistic objective.

In conclusion, this work brings together machine learning-based discriminative network analysis and high-throughput mechanical phenotyping to establish a blueprint workflow for data-driven de novo identification of genes involved in the regulation of cell mechanics. Ultimately, identifying ways to tune the mechanical properties on demand will enable turning cell mechanics from a correlative phenomenological parameter to a controllable property. Such control will, in turn, allow us to interfere with important processes such as tissue morphogenesis, cell migration, or circulation through vasculature.

**Table 5.** Carcinoma cell lines.

List of all carcinoma cell lines acquired from RIKEN BRC Cell Bank used in this study, together with the catalogue number, tissue of origin, carcinoma type, growth medium specification, and passage number at purchase.

| Cell line | Cat no. | Tissue | Type | Medium (Gibco cat #) | Serum (%) | Passage |
|---|---|---|---|---|---|---|
| ECC4 | RCB: RCB0982; RRID:CVCL_1190 | Intestine | small-cell | RPMI1640 (11875093) | 10 | 7 |
| TGBC18TKB | RCB: RCB1169; RRID:CVCL_3338 | Intestine | adeno | DMEM (11885084) | 5 | 5 |
| WA-hT | RCB: RCB2279; RRID:CVCL_8766 | Lung | small-cell | MEM (11095080) | 10 | 54 |
| A549 | RCB: RCB0098; RRID:CVCL_0023 | Lung | adeno | DMEM (11885084) | 10 | 92 |
| ECC10 | RCB: RCB0983; RRID:CVCL_1188 | Stomach | small-cell | RPMI1640 (11875093) | 10 | 8 |
| MKN45 | RCB: RCB1001; RRID:CVCL_0434 | Stomach | adeno | RPMI1640 (11875093) | 10 | 6 |
| MKN1 | RCB: RCB1003; RRID:CVCL_1415 | Stomach | adeno | RPMI1640 (11875093) | 10 | 6 |

# Materials and methods

## Cell culture

### Glioblastoma cell lines

The glioblastoma dataset contained three primary human brain tumor cell lines (X01, X04, and X08) in three distinct signaling states. The cells were cultured and characterized within a framework of a previous study (*Poser et al., 2019*). In brief, the three signaling states characterized by low, medium, and high activation of STAT3-Ser/Hes3 signaling axis, were maintained by growth media containing fetal bovine serum (serum), epidermal growth factor (EGF), or basic fibroblast growth factor combined with a JAK inhibitor (FGFJI), respectively. Upon thawing, cells were expanded in a serum-free DMEM/F12 medium (10-090-CV, Mediatech, Corning, NY, USA) containing N2 supplement and 20 ng ml$^{-1}$ EGF (R&D Systems, MN, USA) at 37°C in a 5% oxygen incubator. Each cell line was then plated into three separate flasks and cultured in the DMEM/F12 medium containing N2 supplement and additional supplementation of either serum (10%), EGF (20 ng ml$^{-1}$), or FGFJI (20 ng ml$^{-1}$, bFGF, R&D Systems; and 200 nM JAK inhibitor, Calbiochem, Merck Millipore, Germany). Cells were collected for mechanical characterization and RNA-Seq after 5-day exposure to the respective culture conditions (*Poser et al., 2019*).

### Carcinoma cell lines

Small-cell and adenocarcinoma cell lines from intestine, stomach, and lung were acquired from RIKEN BioResource Research Center, Japan (see *Table 5* for the list of cell lines and media). Cells were cultured in growth media supplemented with 5% (TGBC) or 10% (rest) heat-inactivated fetal bovine serum (10270106, Gibco, Thermo Fisher Scientific, MA, USA) and 100 U ml$^{-1}$/100 µg ml$^{-1}$ penicillin/streptavidin (15140122, Gibco), at 37°C and 5% $CO_2$. Sub-culturing was performed using trypsin (25200072, Gibco). Cells were collected for mechanical characterization at 70% confluency. The RNA-Seq data was retrieved from FANTOM5 consortium (*Forrest et al., 2014*). Additional transcriptomic datasets were retrieved from the CCLE project microarray (*Barretina et al., 2012*) and RNA-Seq (*Ghandi et al., 2019*) and from the study conducted by Genentech (*Klijn et al., 2015*) (see *Table 1* for overview).

### MCF10A PIK3CA cell lines

MCF10A cell line with single-allele PIK3CA H1024R mutation was previously generated by homologous recombination by Horizon Discovery LTD, UK (*Juvin et al., 2013*) and was kindly provided, together with an isogenic wild-type (wt) control, by L.R. Stephens (Babraham Institute, UK). Cells used for mechanical characterization were cultured in DMEM/F12 medium (31330038, Gibco) supplemented with 5% horse serum (PAA Laboratories), 10 µg ml$^{-1}$ insulin (I9278, Sigma-Aldrich, MO, USA), 0.2 µg ml$^{-1}$ hydrocortisone (H0888, Sigma-Aldrich), 0.1 µg ml$^{-1}$ cholera toxin (C8052, Sigma-Aldrich), and 100 U ml$^{-1}$/100 µg ml$^{-1}$ penicillin/streptomycin (15140122, Gibco). The wt cells were additionally supplemented with 10 ng ml$^{-1}$ EGF (E9644, Sigma-Aldrich), while mutant cell lines were maintained without EGF. Sub-confluent cells were collected for mechanical characterization using trypsin

(25200056, Gibco). Mechanical data were collected from two biological replicates with three technical repetitions each. The RNA-Seq data was retrieved from a previous study (*Kiselev et al., 2015*), in which cells were cultured in a reduced medium (DMEM/F12 supplemented with 1% charcoal dextran treated fetal bovine serum, 0.2 µg ml⁻¹ hydrocortisone, and 0.1 µg ml⁻¹ cholera toxin).

## Induced pluripotent stem cells

F- and C-class iPSCs were derived through reprogramming of murine fetal neural progenitor cells with Tet-On system for doxycycline-inducible expression of OSKM (Oct4, Sox2, Klf4, and cMyc) factors in a previous study (*Urbanska et al., 2017*). Both iPSCs classes were cultured on 0.1% gelatin-coated dishes in FCS/LIF medium [DMEM+Glutamax (61965059, Gibco), 15% fetal calf serum (Pansera ES, PAN-Biotech, Germany), 100 µM β-mercaptoethanol (PAN-Biotech), 2 mM L-glutamine, 1 mM sodium pyruvate, 1× nonessential amino acids, 15 ng ml⁻¹ recombinant LIF (MPI-CBG, Dresden, Germany)]. The F-class iPSCs were additionally supplemented with 1 µg ml⁻¹ doxycycline, and the C-class iPSCs with a mixture of two inhibitors (2i): 1 µM MEK inhibitor (PD0325901, Calbiochem) and 3 µM GSK3 inhibitor (CH99021, Calbiochem). Cells were passaged and harvested using 0.1% trypsin solution. The mechanical characterization was performed not earlier than at the 27th day of reprogramming (*Urbanska et al., 2017*). The microarray expression profiles were retrieved from a previous study, in which the F- and C-class iPSCs were derived from embryonic fibroblasts using similar doxycycline-inducible OSKM expression system (*Tonge et al., 2014*).

## Developing neurons

For isolation of neurons at different developmental stages, we used a double-reporter mouse line *Btg2RFP/Tubb3GFP*, in which the PPs are double negative (RFP−/GFP−), NNs are double positive (RFP+/GFP+), and the cells positive for RFP but negative for GFP (RFP+/GFP−) are the differentiating progenitors that were not used in this study. Lateral cortices dissected from E14.5 murine embryos were dissociated using a papain-based neural dissociation kit (Miltenyi Biotech, Germany) and the cell populations of interest were separated based on the RFP/GFP expression using FACS as described in detail elsewhere (*Aprea et al., 2013*). The three types of sorted cells were then subjected to RNA-Seq (*Aprea et al., 2013*) and mechanical characterization. The animal experiments were approved by the Landesdirektion Sachsen (24-9168.11-1/41 and TVV 39/2015) and carried out in accordance with the relevant guidelines and regulation.

## Mouse embryonic fibroblasts

Previously established, immortalized WT and CAV1KO mouse embryonic fibroblasts derived from WT and CAV1KO littermate C57BL/9 mice (*Razani et al., 2001*) were kindly provided by M.P. Lisanti (University of Salford, Manchester, UK). Cells were cultured in DMEM medium (11960044, Gibco), supplemented with 10% fetal bovine serum (10270106, Gibco), 2 mM glutamine (25030081, Gibco), 100 U ml⁻¹/100 µg ml⁻¹ penicillin/streptomycin (15070063, Gibco), at 37°C and 5% $CO_2$. Sub-confluent cells were collected for mechanical measurements by trypsinization (25200056, Gibco).

## MCF10A-ER-Src cell line

The MCF10A-ER-Src cells were a kind gift from K. Struhl (Harvard Medical School, MA, USA). ER-Src is a fusion of the v-Src (viral non-receptor tyrosine kinase) with the ligand-binding domain of the estrogen receptor, that can be induced by cell treatment with TAM (*Hirsch et al., 2009*). Cells were grown at 37°C under 5% $CO_2$ in DMEM/F12 medium (11039047, Gibco), supplemented with 5% charcoal (C6241, Sigma-Aldrich)-stripped horse serum (16050122, Gibco), 20 ng ml⁻¹ EGF (AF-100–15, Peprotech), 10 mg ml⁻¹ insulin (I9278, Sigma-Aldrich), 0.5 mg ml⁻¹ hydrocortisone (H0888, Sigma-Aldrich), 100 ng ml⁻¹ cholera toxin (C8052, Sigma-Aldrich), and 100 U ml⁻¹/100 µg ml⁻¹ penicillin/streptomycin (15070063, Gibco). To induce the Src expression cells were plated at 50% confluency, and after allowing to adhere for 24 hr, treated with 1 µM 4OH-TAM (H7904, Sigma-Aldrich) or with identical volume of ethanol as a control. Cells were characterized in adherent state using AFM at timepoints specified in the text.

**Table 6.** Mechanical characterizations of cells from the individual datasets using real-time deformability cytometry (RT-DC) — experimental details.

For each dataset, experimental details of the measuring conditions are listed, including the widths of channel constriction ($w_{channel}$), total flow rates ($Q_{total}$), percentages of methylcellulose (MC) in the measurement buffer (buffer % MC), effective viscosity of the measurement buffer in the channel at the flowrate used ($\eta_{eff}$, according to *Herold, 2017*), as well as gates used for data filtering.

| | Measurement conditions | | | | Data filtering | |
|---|---|---|---|---|---|---|
| | $w_{channel}$ (µm) | $Q_{total}$ (µl s$^{-1}$) | Buffer % MC | $\eta_{eff}$ (mPa s$^{-1}$) | Area (µm$^2$) | Area ratio |
| Glioblastoma | 30 | 0.16 | 0.5 | 5.4 | 50–600 | 1.0–1.05 |
| Carcinoma | 30 | 0.16 | 0.5 | 5.4 | 60–600 | 1.0–1.05 |
| MCF10A | 20 | 0.04 | 0.5 | 5.7 | 75–320 | 1.0–1.05 |
| iPSCs | 20 | 0.04 | 0.5 | 5.7 | 50–500 | 1.0–1.05 |
| dev neurons | 20 | 0.04 | 0.5 | 5.7 | 25–300 | 1.0–1.05 |
| MEFs | 30 | 0.16 | 0.5 | 5.4 | 50–500 | 1.0–1.05 |

## Mechanical measurements

### Mechanical characterization of cells using RT-DC

RT-DC measurements for mechanical characterization of cells were performed at room temperature according to previously established procedures (*Urbanska et al., 2018*). In brief, cells were harvested by trypsinization (adherent cells) and/or centrifugation at 400 × *g* for 3–5 min, and suspended in a measurement buffer (MB). MB (osmolarity 310–315 mOsm kg$^{-1}$, pH 7.4) was based on phosphate buffered saline without Mg$^{2+}$ and Ca$^{2+}$ and contained 0.5% or 0.6% (wt/wt) methylcellulose (036718.22; 4000 cPs, Alfa Aesar, Germany) for increased viscosity. Cells were introduced into a microfluidic chip using a syringe pump (NemeSys, Cetoni, Germany), and focused into a 300-µm long channel constriction (with a square cross-section of 20 × 20 or 30 × 30 µm) by sheath flow infused at a flow rate three times as high as that of the cell suspension. The imaging was performed at the end of the channel constriction (*Figure 1—figure supplement 1B*) at 2000 frames s$^{-1}$. The cell area and deformation were derived from the fitted cell contours in real time by the acquisition software (*ShapeIn2*; Zellmechanik Dresden, Germany). Apparent Young's modulus values were assigned to each cell based on its area and deformation under given experimental conditions (flow rate, channel size, viscosity of the medium, and temperature) using a look-up table obtained through numerical simulations of an elastic solid (*Mokbel et al., 2017*) with the aid of *ShapeOut* (*ShapeOut 1.0.10*; available on GitHub; *Müller et al., 2020*). The events were filtered for area ratio (the ratio between the area enclosed by the convex hull of the cell contour and the raw area enclosed by the contour) to discard incomplete contours or cells with rough surface, and for cell area and aspect ratio to discard derbies and doublets. Experimental details (channel sizes, flow rates, and MBs) and gates used for filtration in respective datasets are listed in *Table 6*.

### Mechanical characterization of cells using AFM

For AFM measurements, cells were seeded on glass bottom dishes (FluoroDish; FD35100, WPI, FL, USA) at least 1 day in advance. Mechanical characterization was performed on adherent cells in a subconfluent culture in CO$_2$-independent medium (18045054, Gibco) at 37°C (temperature was maintained by a petri dish heater, JPK Instruments, Germany). AFM measurements on TGBC and ECC4 cell lines were conducted on a *Nanowizard 4* (JPK Instruments/Bruker). Tip-less cantilevers (PNP-TR-TL, nominal spring constant *k* = 0.08 N m$^{-1}$, Nanoworld, Switzerland) decorated a polystyrene bead of 5 µm diameter (PS-R-5.0, microParticles, Germany) each were used as the indenters. The cantilever spring constants were measured prior to each experiment using the thermal noise method implemented in the *JPK SPM* software (JPK Instruments). For each cell, three indentation curves were recorded with a piezo extension speed of 5 µm s$^{-1}$ to a maximum set force of 2 nN. For the microrheology analysis, the cantilever was lowered using a piezo extension speed of 5 µm s$^{-1}$ until a force set point of 1 nN was reached, corresponding to an approximate indentation depth $\delta_0$ of 1 µm. The

lowered cantilever was then oscillated by a sinusoidal motion of the piezo elements at an amplitude of 10 nm for a period of 10 cycles. The oscillations were performed sequentially at different frequencies in the range of 3–200 Hz. Indentation experiments on MCF10A-ER-Src cells were conducted as described above, except different tip-less cantilevers (Arrow TL1, nominal spring constant $k = 0.035$–$0.045$ Nm$^{-1}$, Nanoworld) with a 5 µm bead glued at the end were used as the indenter.

## AFM indentation data analysis

Recorded force–distance curves were converted into force–indentation curves and fitted in JPK data processing software (*JPK DP*, JPK Instruments/Bruker) using Sneddon's modification of the Hertz model for a spherical indenter **Sneddon, 1965**:

$$F = \frac{E}{1 - v^2} \left( \frac{a^2 + r^2}{2} \ln \frac{r + a}{r - a} - ar \right),$$ (3)

With

$$\delta = \frac{a}{2} \ln \frac{r + a}{r - a},$$ (4)

where $F$ denotes the indentation force, $E$ the elastic modulus, $v$ the Poisson's ratio, $a$ the radius of the projected contact area formed between the sample and the indenter, $r$ the radius of the indenter, and $\delta$ the indentation depth. Poisson ratio was set to 0.5. The curves were fitted to a maximal indentation of 1.5 µm.

## AFM microrheology data analysis

The force and indentation signals from oscillatory measurements were fitted using a sinusoidal function to extract the amplitude and phase angle of each signal. Data were analyzed analogously to the procedure described by **Alcaraz et al., 2003** but for a spherical not a pyramidal indenter. Briefly, the method relies on the linearization of the Hertz model for a spherical indenter due to small oscillations by using the first term of the Taylor expansion and subsequent transformation to the frequency domain:

$$F(\omega) = 2 \frac{E^*(\omega)}{(1 - v^2)} \sqrt{R\delta_0} \delta(\omega),$$ (5)

where $F(\omega)$ and $\delta(\omega)$ are the force and indentation signals in the frequency domain, respectively, $E^*(\omega)$ is the complex Young's modulus, $v$ is the Poisson's ratio assumed to be 0.5, $R$ is the radius of the indenter and $\omega$ is the angular frequency. The complex shear modulus $G^*(\omega)$ can be written using $G^*(\omega) = \frac{E^*(\omega)}{2(1+v)}$ :

$$G^*(\omega) = G'(\omega) + iG''(\omega) = \frac{(1 - v)}{4\sqrt{R\delta_0}} \frac{F(\omega)}{\delta(\omega)},$$ (6)

where $G'(\omega)$ is the storage modulus and $G''(\omega)$ is the loss modulus. The ratio of the force $F(\omega)$ and indentation $\delta(\omega)$ is calculated from the measured amplitudes $A^F(\omega)$ and $A^\delta(\omega)$ and the phase shifts $\theta^F(\omega)$ and $\theta^\delta(\omega)$ of the oscillatory signals (**Rother et al., 2014**):

$$\frac{F(\omega)}{\delta(\omega)} = \frac{A^F(\omega)}{A^\delta(\omega)} e^{i\left(\theta^F(\omega) - \theta^\delta(\omega)\right)},$$ (7)

where the difference of the phase shifts $\left(\theta^F(\omega) - \theta^\delta(\omega)\right)$ is in the range of 0° (elastic solid) and 90° (viscous fluid). Furthermore, the hydrodynamic drag contribution to the cantilever oscillation was estimated and subtracted from the complex shear modulus as previously described (**Alcaraz et al., 2002**):

$$G^*(\omega) = \frac{(1 - v)}{4\sqrt{R\delta_0}} \left[ \frac{F(\omega)}{\delta(\omega)} - i\omega b(0) \right],$$ (8)

**Table 7.** siRNAs used in the knock-down experiments.

Full sequences of esiRNAs (HU-03125-1, HU-03125-2, and HU-03125-3) are included in *Supplementary file 5*.

| Name | Target | Commercial name | Cat no. | Vendor |
|---|---|---|---|---|
| rLuc | Renilla Luciferase | RLUC | RLUC | Eupheria Biotec |
| esiCAV1-1 | Human caveolin 1 | hCAV1 | HU-03125-1 | Eupheria Biotec |
| esiCAV1-2 | Human caveolin 1 | hCAV1, custom design | HU-03125-2 | Eupheria Biotec |
| esiCAV1-3 | Human caveolin 1 | hCAV1, custom design | HU-03125-3 | Eupheria Biotec |
| nonT | Non-targeting | ON-TARGETplus Non-targeting Pool | D-001810-10-05 | Dharmacon |
| CAV1-pool | Human caveolin 1 | ON-TARGETplus Human CAV1 siRNA, SMARTPool | L-003467-00-0005 | Dharmacon |

where $b(h)$ is the hydrodynamic drag coefficient function measured from non-contact oscillations of the cantilever at different distances $h$ from the sample, and $b(0)$ is the extrapolation to distance 0 from the sample. For PNP-TR-TL cantilevers, the hydrodynamic drag coefficient was estimated to be $b(0) = 5.28\,\mu\mathrm{N\,s\,m}^{-1}$.

## Perturbation experiments

### CAV1 knock-down

For RNAi experiments, cells were transfected using RNAiMax reagent (13778030, Thermo Fisher Scientific) and a reverse transfection protocol. Per transfection, 200 ng of esiRNA (Eupheria Biotech, Germany) or 300 ng of ON-TARGETplus siRNA (Dharmacon, CO, USA) and 2 µlRNAiMax were prepared in OptiMEM (31985062, Gibco) according to the manufacturer's instructions and pipetted onto 12-well plates (see *Table 7* for full list of siRNAs used). Cells in 1 ml of culture medium were plated on top of the transfection mix at a density allowing for sub-confluent growth within the experimental timeframe. Seventy-two hours post transfection, cells were collected for the mechanical characterization and western blot analysis.

### Plasmid for CAV1 overexpression

The cDNA of *CAV1* was amplified by PCR, introducing NheI and XhoI restriction sites in the flanking regions. The PCR product was then cloned into the pCGIT destination vector (a kind gift from P. Serup, University of Copenhagen, Denmark) under the CAG promoter and with dTomato fluorescent marker under internal ribosomal entry site (IRES) downstream of CAV1. The pCGIT0-hCAV1 plasmid map together with the pCGIT destination vector map are available on figshare.

### Transient CAV1 overexpression in ECC4 and TGBC cells

ECC4 and TGBC cells were transiently transfected with the CAV1 overexpression plasmid by electroporation (Neon Transfection System, MPK5000, Thermo Fisher Scientific). Per transfection 0.3 × 10⁶ ECC4 cells, or 0.2 × 10⁶ TGBC cells were mixed with 1 µg of plasmid DNA in PBS. Electroporation was conducted using 10 µl Neon tips (MPK1096, Thermo Fisher Scientific) and a program of two pulses of 1050 V and 30ms duration each. Electroporated cells were transferred to 500 µl of fresh culture medium in a 24-well plate. The cells were collected for mechanical characterization and western blot analysis 72 hr post transfection. To identify fluorescent cells during mechanical characterization, the combined RT-FDC (*Rosendahl et al., 2018*) setup was used, and the maximum intensity of the fluorescence signal from channel 2 (excitation 561 nm, 10% laser power; collection 593/46) was utilized for gating.

### Transient CAV1 overexpression in MCF10A-ER-Src cells

MCF10A-ER-Src cells were transiently transfected with the CAV1 overexpressing plasmid using Effectene transfection reagent (301425, QIAGEN). One day before transfection, cells were seeded on glass bottom 35 mm dishes (FluoroDish; FD35100, WPI, FL, USA) at a density of 20,000 cells per well. Transfection was performed according to the manufacturer's instruction using 75 µl EC buffer, 0.6 µg plasmid DNA, 4.8 µl Enhancer, and 6 µl Effectene reagent per well. Twenty-four hours post

transfection cells were induced with 1 µM TAM. Mechanical analysis was performed after additional 72 hr of culture.

## Western blotting

For western blot analysis of carcinoma and MCF10A-ER-Src cell lines, cell pellets were collected in parallel with mechanical measurements and lysed using ice-cold RIPA buffer (89900, Thermo Fisher Scientific) supplemented with protease/phosphatase inhibitor cocktail (78441, Thermo Fisher Scientific) and benzonase (E1014, Sigma-Aldrich). The lysates were cleared at 4°C by 10-min sonication followed by 10-min centrifugation at 16,900 × g. Obtained supernatants were mixed with Laemmli buffer (final concertation: 62.5 mM Tris–HCl pH 6.8, 2% SDS, 10% glycerol, 5% β-mercaptoethanol, and 0.01% bromophenol blue), boiled (5 min at 95°C), and separated by SDS–PAGE electrophoresis on 4–20% gradient gels (Mini-PROTEAN TGX Precast Gels; 4561093, Biorad, CA, USA) in MOPS SDS Running buffer (B0001, Thermo Fisher Scientific). After transferring the proteins onto a PVDF membrane (Merck Millipore), the membranes were blocked in TBS-T (20 mM Tris, 137 mM NaCl, 0.1% Tween) containing 5% wt/vol skimmed milk powder (T145.1, Carl Roth, Germany) for 40 min. Next, membranes were incubated with the primary anti-Cav1 (1:1000; D46G3; #3267, Cell Signaling Technology, MA, USA) and anti-GAPDH (1:5000; ab9485, Abcam, UK) antibodies at 4°C overnight in 5% milk/TBS-T, washed, and incubated with anti-rabbit HRP-conjugated secondary antibody (1:4000; ab97069, Abcam). Chemiluminescence detection was performed using Pierce Enhanced Chemi-Luminescence (ECL) substrate (32109, Thermo Fisher Scientific) and ECL films (GE28-9068-37, Merck Millipore). Films were developed in an OptiMax X-ray film processor (KODAK, NY, USA). Quantitative analysis was performed on scanned films using the gel analysis tool in *Fiji/JmageJ* version 2.0.0-rc-69/1.52p (https://fiji.sc/, *Schindelin et al., 2012*). For western blot analysis of MEFs the same anti-Cav1 antibody (1:1000; D46G3; #3267, Cell Signaling) was used, and anti-tubulin antibody (1:2000; DM1A; #3873, Cell Signaling) was used as a loading control. Goat anti-mouse 680 and goat anti-rabbit 800 (1:2000; A28183 and A32735, Thermo Fisher Scientific) antibodies were used for secondary detection. Membranes were scanned with the Odyssey imaging system (LI-COR Biosciences, NE, USA).

## Computational analysis

### Transcriptomic datasets

Transcriptomic datasets were retrieved from online databases (Gene Expression Omnibus, GEO and DNA Data Bank of Japan, DDBJ) with accession numbers listed in *Table 1*. An overview of experimental details for RNA profiling procedures and data analysis in individual datasets is presented in *Supplementary file 2*. The IDs of samples used in respective categories in each dataset are listed in *Supplementary file 3*. In case of multiple entries for the same gene in a given transcriptomic dataset, the expression values were averaged, so that only one entry per gene and sample was available.

### PC-corr analysis

Before performing the PC-corr analysis, the glioblastoma and iPSC datasets were intersected and normalized by taking the log10 (glioblastoma dataset) or *z*-score (iPSC dataset) of the subset of 9452 overlapping genes. The PC-corr analysis was conducted on individual datasets as described in detail elsewhere (*Ciucci et al., 2017*). In brief, PCA was performed using *svd* function in *MATLAB* (*R2020a*, MathWorks, MA, USA) on normalized datasets. The original PC loadings from the component providing good separation of sample categories (PC1 for both analyzed datasets) were processed in a two-step procedure including the normalization and scaling. The processing of the PC loadings is performed to adjust the distribution of the loadings to the range of Pearson's correlation values [−1,1], so that they are comparable when computing the *PC-corr* value. The normalization was performed using a custom function developed previously (*Ciucci et al., 2017*) of the following formula:

$$V_i^* = \text{sgn}\left(V_i^0\right) \log_{10}\left(1 + \frac{\left|V_i^0\right|}{\langle|V^0|\rangle}\right), \tag{9}$$

where $V_i^*$ denotes the normalized loading corresponding to the $i$th feauture, $V_i^0$ the original loading corresponding to the $i$th feauture, and $\langle |V^0| \rangle$ the average of all absolute loadings of the vector $V^0$.

The normalized loadings were then scaled to fall on the interval $[-1,1]$ using a previously developed custom function (*Ciucci et al., 2017*):

$$V_i = \mathrm{sgn}\left(V_i^*\right) \frac{|V_i^*| - \min\left(|V^*|\right)}{\max\left(|V^*|\right) - \min\left(|V^*|\right)}, \tag{10}$$

where $V_i$ denotes the processed loading corresponding to the $i$th feature, and $|V^*|$ the vector containing absolute values of all normalized loadings.

The *PC-corr* values for each pair of features were computed according to *Equation 1*. The *PC-corr* results of the glioblastoma and iPSC datasets were combined as described in the results section. Gene pairs showing different PC-corr signs were masked by setting the *PC-corr$^{comb}$* to zero. The genes and edges comprising the network were obtained via thresholding strategies described in the main text. The network was visualized using *cytoscape* (*cytoscape 3.8.0*; https://cytoscape.org/) (*Shannon et al., 2003*).

## Combinatorial marker

To compute a combinatorial marker associated with a gene functional network module composed of $n$ genes, we use the following three-step procedure.

### Step 1: Dataset normalization

To scale the features to a comparable range and reduce the dominant influence of highly expressed genes, each dataset is normalized. Possible normalization approaches include logarithm normalization ($x = \log(x+1)$) and $z$-score normalization. Since both normalization approaches lead to comparable results, we decided to proceed with the logarithm normalization because it is one of the most widely adopted in computational genomics for combinatorial markers (*Danaher et al., 2017*).

### Step 2: Direction alignment

The direction of the gene expression change between different samples is analyzed and, if necessary, aligned. The pairwise Pearson's correlation of the $n$ genes from the dataset used for inference of the combinatorial marker is computed. If all pairs of genes are positively correlated between each other there is no need of direction alignment — this was the specific case in our study. Otherwise, the directions of genes whose correlation with the reference gene is negative need to be aligned before the compression. The reference gene for the direction alignment is the gene with the highest average pairwise Pearson's correlation with the other $n-1$ genes in the functional module. The alignment is performed by subtracting the mean value of the normalized expression across samples, $\bar{g}$, from the normalized expression of the given gene, $g$, inverting its trend using the multiplication by $-1$, and finally adding again the mean value to regain the original expression level:

$$\mathrm{align}\left(g\right) = -\left(g - \bar{g}\right) + \bar{g} = 2\bar{g} - g. \tag{11}$$

Once defined, the aligned values should be used for any further validation analysis, including the computation of the JVT. The alignment step is necessary to make sure that the information contained in the anticorrelated genes does not annihilate each other during the compression into the combinatorial marker. Below, an example is provided to illustrate this issue.

### Step 3: Compression

To perform compression and obtain the combinatorial marker $g_{comb}$ we employ one of the most employed compression operators in computational genomics (*Danaher et al., 2017*), the mean operator:

$$g_{comb} = \frac{1}{n}\sum_{i=1}^{n} g_i, \tag{12}$$

where $g_i$ indicates the normalized and aligned expression value of the $i$th gene of the functional module from which the combinatorial marker is derived.

To illustrate the importance of the alignment, let us consider a simple example of two anticorrelated genes in four samples: $g_1$ = [1 1 3 3] and $g_2$ = [3 3 1 1]. When the compression is performed without alignment, following values of the combinatorial marker are obtained: $g_{comb} = \frac{g_1 + g_2}{2}$ = [2 2 2 2]. The so-obtained combinatorial marker is non-discriminative, even though the individual genes are. On the contrary, if the alignment function is applied prior to compression:
$g_{comb} = \frac{g_1 + \text{align}(g_2)}{2} = \frac{[1133] + [4 - [3311]]}{2} = \frac{[1133] + [1133]}{2} = [1133]$, the original discriminative information is conserved in the combined marker.

## Joint-view trustworthiness

The single-view trustworthiness measure was recently introduced by us in studies on pattern recognition to assess the extent to which the geometrical discrimination of samples of a dataset might emerge at random along a dimension of embedding in a geometrical space (*Durán et al., 2021*; *Acevedo et al., 2022*). In brief, the single-view trustworthiness measure is an empirical p value computed from a null model distribution obtained by a resampling technique, which randomly shuffles the labels of the samples and computes what is the probability to generate at random a matching between sample labels and sample geometrical location that offers a discrimination that is equal or larger than the one tested. The obtained p value assesses whether the visualized and measured sample discrimination along a dimension of a geometrical space is significant (because rare to appear at random) or no significant (because frequent to appear at random). This is particularly useful to assess the trustworthiness of a discriminative result when the number of samples for each class is small or when it is unbalanced, as is the case for some datasets in our study. To assess the trustworthiness of a marker's discrimination performance jointly on many datasets, we introduce a joint-view extension to this method which we refer to as the JVT.

To ensure that the proposed markers have a joint multiview discrimination that is rare to obtain by chance, JVT samples markers at random from the data and compares their performance to the one of predicted targets according to the following procedure:

1. *Data preparation*: Collect datasets that support (positive hypothesis: for instance, discriminative presence of a cell mechanic phenotype: soft/stiff) or not support (negative hypothesis: for instance, discriminative absence of a cell mechanic phenotype: soft/stiff) your hypothesis, and make sure that you consider for all of them only the features (genes) that are common to each dataset in you study.
2. *Data normalization*: Perform only when computing combinatorial marker (see Step 1 in the Combinatorial marker section above).
3. *Null model distribution sampling and p-value estimation*: (a) *Single marker test*: Sample at random a gene and extract its expression from each dataset, compute its joint multiview discrimination performance as the minimum performance measure (we adopted the AUC-ROC because it is one of the most used in classification assessment, but any classification performance measure can be employed) across the datasets. Repeat this procedure sampling at random for $T$ times (in our study we used $T$ = 10,000) a gene from the datasets and computing its minimum classification performance measure across the datasets. The ensemble of the $T$ minimum classification performance measures can be used to draw an empirical distribution that forms the null model. The p value of the tested marker is computed counting the proportion of genes that within the $T$ samplings have a minimum classification performance that is equal or larger than that for the tested marker. Please note that here we compute the joint multiview discrimination performance using the minimum performance across the datasets because we pursue a conservative estimation. Other operators, such as mean, median, or mode can be employed instead of the minimum operator to make the JVT estimation less conservative. (b) *Combinatorial marker test*: Given a combinatorial marker of $m$ genes, sample at random $m$ genes and extract their expressions from each dataset, compute the combinatorial marker (apply the same compression formula of the tested combinatorial marker) joint multiview discrimination performance as the minimum performance measure we adopted the AUC-ROC, see point (3a) for details across the datasets. Repeat this procedure $T$ times (in our study we used $T$ = 10,000). The p value is computed as for the single marker test (see point 3a).

The JVT pseudocode and time complexity analysis are provided in *Supplementary file 4*. In brief, the overall complexity of JVT considering a scenario like in our study is O(Z+T), that is, JVT is linear in $Z$ (number of common genes in the datasets) and $T$ (number of samplings). The JVT code (in MATLAB, R, and Python) and datasets to replicate the results in *Table 4* of this study are available on GitHub (copy archived at *biomedical-cybernetics, 2022*).

## Statistical analysis

The RT-DC datasets were compared using generalized linear mixed effects models with the aid of ShapeOut (ShapeOut 1.0.1; available on GitHub; *Müller et al., 2020*) as described in detail elsewhere (*Herbig et al., 2018*). AFM datasets were compared using two-sided Wilcoxon rank sum test in MATLAB (R2020a, MathWorks). Western blot results were compared using a two-sided two-sample $t$-test in MATLAB (R2020a, MathWorks).

## Acknowledgements

We thank Isabel Richter and Christine Schweitzer for technical assistance, Miguel Sanchez (CNIC, Spain) and Konstantinos Anastasiadis (TU Dresden, Germany) for helpful discussions, Len R Stephens (Babraham Institute, UK) for provision of MCF10A PIK3CA cells, and Kevin Struhl (Harvard Medical School, MA, USA) for provision of MCF10A-ER-Src cells. We further thank the Microstructure Facility at the Center for Molecular and Cellular Bioengineering (CMCB) at the Technische Universität Dresden (in part funded by the State of Saxony and the European Regional Development Fund) for hosting the chip fabrication. The authors acknowledge the following funding: Alexander von Humboldt-Stiftung, Alexander von Humboldt Professorship (JG), European Commission, ERC Starting Grant 'LightTouch' #282060 (JG), Marie Sklodowska-Curie Actions under the European Union's Horizon 2020 research and innovation programme, BIOPOL ITN, #641639 (MADP, JG), Deutsche Forschungsgemeinschaft, #GU 612/5-1 and #399422891 (JG), Zhou Yahui Chair Professorship of Tsinghua University (CVC), The starting funding of the Tsinghua Laboratory of Brain and Intelligence (THBI) (CVC), The National High-Level Talent Program of the Ministry of Science and Technology of China #20241710001 (CVC), The independent research group leader running funding of the Technische Universität Dresden (CVC), Wellcome Trust, Sir Henry Wellcome Postdoctoral Fellowship, #224074/Z/21/Z (MU), Comunidad Autónoma de Madrid, Tec4Bio-CM, #S2018/NMT-4443 (MADP), Fundació La Marató de TV3, #201936-30-31 (MADP), Mildred Scheel Early Career Center Dresden (MSNZ) funded by the German Cancer Aid (Deutsche Krebshilfe) (AT).

## Additional information

### Competing interests

Shada Abuhattum, Martin Kräter, Jochen Guck: Co-founder and shareholder of the company Rivercyte GmbH that is commercializing deformability cytometry technology. The other authors declare that no competing interests exist.

### Funding

| Funder | Grant reference number | Author |
| --- | --- | --- |
| Alexander von Humboldt-Stiftung | Alexander von Humboldt Professorship | Jochen Guck |
| European Commission | ERC Starting Grant "LightTouch" 282060 | Jochen Guck |
| European Commission | 641639 | Miguel Ángel del Pozo Jochen Guck |
| Deutsche Forschungsgemeinschaft | GU 612/5-1 | Jochen Guck |
| Deutsche Forschungsgemeinschaft | 399422891 | Jochen Guck |

| Funder | Grant reference number | Author |
| --- | --- | --- |
| Comunidad de Madrid | S2018/NMT-4443 | Miguel Ángel del Pozo |
| Fundació la Marató de TV3 | 201936-30-31 | Miguel Ángel del Pozo |
| Deutsche Krebshilfe | | Anna Taubenberger |
| Ministry of Science and Technology of the People's Republic of China | 20241710001 | Carlo Vittorio Cannistraci |
| Tsinghua University | Starting Fund | Carlo Vittorio Cannistraci |
| Tsinghua University | Zhou Yahui Chair Professorship | Carlo Vittorio Cannistraci |
| Wellcome Trust | 10.35802/224074 | Marta Urbanska |

The funders had no role in study design, data collection and interpretation, or the decision to submit the work for publication. For the purpose of Open Access, the authors have applied a CC BY public copyright license to any Author Accepted Manuscript version arising from this submission. Open access funding provided by Max Planck Society.

## Author contributions

Marta Urbanska, Data curation, Software, Formal analysis, Funding acquisition, Investigation, Visualization, Methodology, Writing – original draft, Project administration, Writing – review and editing, Performed the mechanical measurements of cells (unless indicated otherwise), Analyzed the experimental data, Assisted with data curation and transcriptomic-based computational analysis, Visualized the data and prepared figures, Prepared the initial version of the manuscript; Yan Ge, Data curation, Software, Formal analysis, Validation, Investigation, Methodology, Writing – review and editing, Under supervision of CVC performed the core of data curation and computational analysis on transcriptomics datasets; Maria Winzi, Data curation, Formal analysis, Investigation, Project administration, Writing – review and editing, Performed and analyzed the MCF10A-ER-Src experiments, Assisted with data curation and project administration, Designed and prepared plasmids for CAV1 over-expression analysis; Shada Abuhattum, Investigation, Methodology, Writing – review and editing, Provided methodological support with AFM measurements and data analysis for TGBC and ECC4 cell lines; Syed Shafat Ali, Software, Formal analysis, Validation, Methodology, Writing – review and editing, Implemented JVT code for in silico validation in python/R and executed the validation analysis; Maik Herbig, Investigation, Writing – review and editing, Performed the mechanical characterisation of developing neurons isolated from mouse embryos and glioblastoma cells; Martin Kräter, Investigation, Writing – review and editing, Performed the mechanical characterisation of human hematopoietic stem cells that were part of the original version of the manuscript; Nicole Toepfner, Investigation, Writing – review and editing, Performed the mechanical characterisation of MCF10A wt/H1047R cells; Joanne Durgan, Resources, Investigation, Writing – review and editing, Provided the cultures of MCF10A wt/H1047R cells; Oliver Florey, Resources, Investigation, Writing – review and editing, Provided the cultures of MCF10A wt/H1047R cells; Martina Dori, Resources, Investigation, Writing – review and editing, Isolated the developing neurons from mouse embryos; Federico Calegari, Resources, Supervision, Writing – review and editing, Provided supervision for the isolation of the developing neurons from mouse embryos; Fidel-Nicolás Lolo, Investigation, Writing – review and editing, Performed MEF CAV1KO experiments; Miguel Ángel del Pozo, Resources, Supervision, Investigation, Writing – review and editing, Povided MEF CAV1KO cells and supervised the experiments with this cells; Anna Taubenberger, Supervision, Investigation, Methodology, Writing – review and editing, Provided methodological support with AFM measurements and data analysis for MCF10A-ER-Src project, as well as advise and conceptual contributions to this manuscript; Carlo Vittorio Cannistraci, Conceptualization, Software, Formal analysis, Supervision, Funding acquisition, Validation, Investigation, Methodology, Project administration, Writing – review and editing, Co-conceived the project, Supervised and developed the codes for the core computational analysis of transcriptomic data presented in this manuscript; Jochen Guck, Conceptualization, Resources, Supervision, Funding acquisition, Project administration, Writing – review and editing, Co-conceived the project; as well as provided conceptual guidance and supervision throughout the project

**Author ORCIDs**
Marta Urbanska (iD) https://orcid.org/0000-0002-6517-5958
Martin Kräter (iD) https://orcid.org/0000-0001-7122-7331
Oliver Florey (iD) https://orcid.org/0000-0002-1075-7424
Federico Calegari (iD) https://orcid.org/0000-0002-3703-2802
Fidel-Nicolás Lolo (iD) https://orcid.org/0000-0003-1635-4770
Miguel Ángel del Pozo (iD) https://orcid.org/0000-0001-9077-391X
Carlo Vittorio Cannistraci (iD) https://orcid.org/0000-0003-0100-8410
Jochen Guck (iD) https://orcid.org/0000-0002-1453-6119

**Ethics**
The animal experiments (isolation of developing neurons from mouse embryos) were approved by the Landesdirektion Sachsen (24-9168.11-1/41 and TVV 39/2015) and carried out in accordance with the relevant guidelines and regulation.

Reviewer #1 (Public review): https://doi.org/10.7554/eLife.87930.3.sa1
Author response https://doi.org/10.7554/eLife.87930.3.sa2

# Additional files

## Supplementary files
Supplementary file 1. Operation parameters of the three methods used for characterizing the mechanical properties of cells.

Supplementary file 2. Overview of transcriptomic profiling details for the datasets used in this study.

Supplementary file 3. List of sample IDs assigned to the different cell states in the respective transcriptomic datasets.

Supplementary file 4. Joint-view trustworthiness (JVT) pseudocode and computational complexity analysis.

Supplementary file 5. Sequences of esiRNAs used for CAV1 knock-down experiments.

MDAR checklist

## Data availability
The transcriptomic data used in this study were obtained from public repositories, their accession numbers are listed in *Table 1*. The mechanical characterization data are available as a collection on figshare. The MATLAB code for performing the PC- corr analysis was based on the code deposited alongside a previous publication (*Ciucci et al., 2017*), accessible on GitHub (*biomedical-cybernetics, 2017*). The JVT code (in MATLAB, R, and Pythonn) and datasets for replicating the results presented in *Table 4* are available on GitHub (copy archived at *biomedical-cybernetics, 2022*) and figshare.

The following datasets were generated:

| Author(s) | Year | Dataset title | Dataset URL | Database and Identifier |
|---|---|---|---|---|
| Urbanska M, Ge Y, Winzi M, Abuhattum S, Herbig M, Ali SS, Herbig M | 2025 | Mechanomics | https://doi.org/10.6084/m9.figshare.c.5399826 | figshare, 10.6084/m9.figshare.c.5399826 |
| Cannistraci CV, Ge Y, Ali SS, Urbanska M | 2025 | Mechanomics Code - JVT | https://doi.org/10.6084/m9.figshare.20123159 | figshare, 10.6084/m9.figshare.20123159 |

The following previously published datasets were used:

| Author(s) | Year | Dataset title | Dataset URL | Database and Identifier |
|---|---|---|---|---|
| Poser S, Lesche M, Dahl A, Ge Y, Cannistraci C | 2019 | Glioblastoma multiforme cancer stem cells from different patients exhibit consistent gene expression and mechanical phenotypes across distinct states in culture | https://www.ncbi.nlm.nih.gov/geo/query/acc.cgi?acc=GSE77751 | NCBI Gene Expression Omnibus, GSE77751 |
| FANTOM5 consortium | 2013 | FANTOM5 CAGE profiles of human and mouse samples | https://ddbj.nig.ac.jp/search/entry/sra-submission/DRA000991 | DNA Data Bank of Japan, DRA000991 |
| Barretina J, Caponigro G, Stransky N, Venkatesan K | 2012 | SNP and Expression data from the Cancer Cell Line Encyclopedia (CCLE) | https://www.ncbi.nlm.nih.gov/geo/query/acc.cgi?acc=GSE36139 | NCBI Gene Expression Omnibus, GSE36139 |
| Broad DepMap | 2021 | DepMap 21Q4 Public | https://doi.org/10.6084/m9.figshare.16924132 | figshare, 10.6084/m9.figshare.16924132 |
| Institute European Bioinformatics | 2011 | [E-MTAB-513] Illumina Human Body Map 2.0 Project | https://www.ncbi.nlm.nih.gov/geo/query/acc.cgi?acc=GSE30611 | NCBI Gene Expression Omnibus, GSE30611 |
| Kiselev VY, Juvin V, Malek M, Luscombe N, Hawkins P, Le Novère N, Stephens L | 2015 | Perturbations of PIP3 signaling trigger a global remodeling of mRNA landscape and reveal a transcriptional feedback loop | https://www.ncbi.nlm.nih.gov/geo/query/acc.cgi?acc=GSE69822 | NCBI Gene Expression Omnibus, GSE69822 |
| Nagy A, Tonge PD | 2014 | Genome-wide analysis of gene expression during somatic cell reprogramming | https://www.ncbi.nlm.nih.gov/geo/query/acc.cgi?acc=GSE49940 | NCBI Gene Expression Omnibus, GSE49940 |
| Aprea J, Prenninger S, Dori M, Sebastian Monasor L, Wessendorf E, Zocher S, Massalini S, Ghosh T, Alexopoulou D, Lesche M, Dahl A, Groszer M, Hiller M, Calegari F | 2013 | Transcriptome Sequencing During Mouse Brain Development Identifies Long Non-Coding RNAs Functionally Involved in Neurogenic Commitment | https://www.ncbi.nlm.nih.gov/geo/query/acc.cgi?acc=GSE51606 | NCBI Gene Expression Omnibus, GSE51606 |

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

# Appendix 1

## Appendix 1—key resources table

| Reagent type (species) or resource | Designation | Source or reference | Identifiers | Additional information |
|---|---|---|---|---|
| Gene (*Homo sapiens*, *Mus musculus*) | CAV1 | NA | HGNC:1527; MGI:102709 | Caveolin 1 |
| Gene (*H. sapiens*, *M. musculus*) | FHL2 | NA | HGNC:3703; MGI:1338762 | Four and a half LIM domains 2 |
| Gene (*H. sapiens*, *M. musculus*) | IGFBP7 | NA | HGNC:5476; MGI:1352480 | Insulin-like growth factor-binding protein 7 |
| Gene (*H. sapiens*, *M. musculus*) | TAGLN | NA | HGNC:11553; MGI:106012 | Transgelin |
| Gene (*H. sapiens*, *M. musculus*) | THBS1 | NA | HGNC:11785; MGI:98737 | Thrombospondin 1 |
| Antibody | anti-Caveolin-1 (rabbit monoclonal) | Cell Signaling Technology | CST: 3267; RRID:AB_2275453 | WB (1:1000) |
| Antibody | anti-GAPDH (rabbit polyclonal) | Abcam | Abcam: ab9485; RRID:AB_307275 | WB (1:5000) |
| Antibody | anti-rabbit HRP-conjugated (goat polyclonal) | Abcam | Abcam: ab97069; RRID:AB_10679812 | WB (1:4000) |
| Cell line (*H. sapiens*) | Glioblastoma | *Poser et al., 2019* | X01; X04; X08 | Human brain tumor cell lines; maintained in A. Androutsellis-Theotokis Lab (TU Dresden, Germany) |
| Cell line (*H. sapiens*) | ECC4 | RIKEN BRC Cell Bank | RCB: RCB0982; RRID:CVCL_1190 | Intestine small-cell carcinoma; passage 7; medium: RPMI1640 (#11875093), 10% FBS |
| Cell line (*H. sapiens*) | TGBC (TGBC18TKB) | RIKEN BRC Cell Bank | RCB: RCB1169; RRID:CVCL_3338 | Intestine adenocarcinoma; passage 5; medium: DMEM (#11885084), 5% FBS |
| Cell line (*H. sapiens*) | WA-hT | RIKEN BRC Cell Bank | RCB: RCB2279; RRID:CVCL_8766 | Lung small-cell carcinoma; passage 54; medium: MEM (#11095080), 10% FBS |
| Cell line (*H. sapiens*) | A549 | RIKEN BRC Cell Bank | RCB: RCB0098; RRID:CVCL_0023 | Lung adenocarcinoma; passage 92; medium: DMEM (#11885084), 10% FBS |
| Cell line (*H. sapiens*) | ECC10 | RIKEN BRC Cell Bank | RCB:RCB0983; RRID:CVCL_1188 | Stomach small-cell carcinoma; passage 8; medium: RPMI1640 (#11875093), 10% FBS |
| Cell line (*H. sapiens*) | MKN45 | RIKEN BRC Cell Bank | RCB: RCB1001; RRID:CVCL_0434 | Stomach adenocarcinoma; passage 6; medium: RPMI1640 (#11875093), 10% FBS |
| Cell line (*H. sapiens*) | MKN1 | RIKEN BRC Cell Bank | RCB: RCB1003; RRID:CVCL_1415 | Stomach adenocarcinoma; passage 6; medium: RPMI1640 (#11875093), 10% FBS |
| Cell line (*H. sapiens*) | MCF10A H1024R; MCF10A WT | *Juvin et al., 2013* | MCF10A H1024R; MCF10A WT | Breast epithelial cells bearing single-allele oncogenic mutation of PIK3CA (H1024R); WT – isogenic control; kindly provided by L.R. Stephens (Babraham Institute, UK) |
| Cell line (*H. sapiens*) | MCF10A-ER-Src | *Hirsch et al., 2009* | MCF10A-ER-Src; RRID:CVCL_N805 | Breast epithelial cell model of TAM-inducible cancerous transformation driven by v-Src, a kind gift from K. Struhl (Harvard Medical School, MA, USA) |
| Cell line (*M. musculus*) | iPSCs (F- and C-class) | *Urbanska et al., 2017* | iPSCs (F- and C-class) | Induced pluripotent stem cells derived through reprogramming of murine fetal neural progenitor cells |
| Cell line (*M. musculus*) | MEFs CAV1KO; MEFs WT | *Razani et al., 2001* | MEFs CAV1KO; MEFs WT | Mouse embryonic fibroblasts derived from WT or CAV1KO littermate C57BL/9 mice; cell lines were a kind gift from M.P. Lisanti (University of Salford, Manchester, UK) |

*Appendix 1 Continued on next page*

*Appendix 1 Continued*

| Reagent type (species) or resource | Designation | Source or reference | Identifiers | Additional information |
|---|---|---|---|---|
| Biological sample (*M. musculus*) | Developing neurons (primary cells) | *Aprea et al., 2013* | Developing neurons: PP – proliferating progenitors; NNs – newborn neurons | Freshly isolated from double-reporter mouse line Btg2$^{RFP}$/Tubb3$^{GFP}$ by M. Dori in the Lab of F. Calegari |
| Transfected construct (*H. sapiens*) | rLuc (esiRNA to rLuc) | Eupheria Biotech | Eupheria Biotech: RLUC | 200 ng per 2 µl RNAiMax in 12wp format |
| Transfected construct (*H. sapiens*) | esiCAV1-1 (esiRNA to human CAV1, design 1, commercially available) | Eupheria Biotech | Eupheria Biotech: HU-03125-1 | 200 ng per 2 µl RNAiMax in 12wp format; see *Supplementary file 5* for sequence details |
| Transfected construct (*H. sapiens*) | esiCAV1-2 (esiRNA to human CAV1, design 2, custom) | Eupheria Biotech | Eupheria Biotech: HU-03125-2 | 200 ng per 2 µl RNAiMax in 12wp format; see *Supplementary file 5* for sequence details |
| Transfected construct (*H. sapiens*) | esiCAV1-3 (esiRNA to human CAV1, design 3, custom) | Eupheria Biotech | Eupheria Biotech: HU-03125-3 | 200 ng per 2 µl RNAiMax in 12wp format; see *Supplementary file 5* for sequence details |
| Transfected construct (*H. sapiens*) | nonT (ON-TARGETplus Non-targeting siRNA Pool) | Dharmacon | Dharmacon: D-001810-10-05 | 300 ng per 2 µl RNAiMax in 12wp format |
| Transfected construct (*H. sapiens*) | CAV1-pool (ON-TARGETplus Human CAV1 siRNA, SMARTPool) | Dharmacon | Dharmacon: L-003467-00-0005 | 300 ng per 2 µl RNAiMax in 12wp format |
| Transfected construct (*H. sapiens*) | pCGIT-hCAV1 (plasmid, plasmid product referred to as CAV1iT) | this paper | pCGIT-hCAV1 | See 'Plasmid for CAV1 overexpression' in Materials and methods; plasmid map available on figshare |
| Chemical compound, drug | RNAiMax reagent | Thermo Fisher Scientific | Thermo Fisher Scientific: 13778030 | For siRNA transfections |
| Chemical compound, drug | Effectene transfection reagent | QIAGEN | QIAGEN: 301425 | For plasmid transfections |
| Chemical compound, drug | Methylcellulose | Alpha Aesar | Cat#: 036718.22 CAS 9004-67-5 | For preparation of viscosity-adjusted RT-DC measurement buffer |
| Software, algorithm | ShapeOut (v 1.0.10) | *Müller et al., 2020* | | For analysis of RT-DC data, available on GitHub |
| Software, algorithm | JPK data processing software | JPK Instruments/Bruker | | For analysis of AFM experiments |
| Software, algorithm | PC-Corr network analysis | *Ciucci et al., 2017* | | Code available on GitHub |
| Software, algorithm | Cytoscape (v 3.8.0) | *Shannon et al., 2003* | RRID:SCR_003032 | https://cytoscape.org/ |
| Software, algorithm | Joint-view trustworthiness (JVT) | this paper; *biomedical-cybernetics, 2022* | | Code available on GitHub and figshare |
| Software, algorithm | Fiji, ImageJ | *Schindelin et al., 2012* | RRID:SCR_002285 | https://fiji.sc/ |
| Other | PNP-TR-TL | Nanoworld, Switzerland | Nanoworld: PNP-TR-TL | Tip-less AFM cantilevers, nominal spring constant $k = 0.08$ N m$^{-1}$ |
| Other | Arrow TL1 | Nanoworld, Switzerland | Nanoworld: Arrow TL1 | Tip-less AFM cantilevers, nominal spring constant $k = 0.035$–$0.045$ N m$^{-1}$ |
| Other | Polystyrene beads, 5 µm diameter | microParticles, Germany | microParticles: PS-R-5.0 | For decorating of the AFM cantilevers |

