## [Editor Report · eLife Assessment]

This **important** study uses machine learning-based network analysis on transcriptomic data from different tissue cell types to identify a small set of conserved (pan-tissue) genes associated with changes in cell mechanics. The new method, which provides a new type of approach for mechanobiology, is accessible, **compelling**, and well-validated using in silico and experimental approaches. The study provides motivation for researchers to test hypotheses concerning the identified five-gene network, and the method will be strengthened over time with expanded sets of validations, such as testing genes with hitherto unknown roles and different perturbation techniques.

---

## [Referee Report · Reviewer #1 (Public review)]

In this work, Urbanska and colleagues use a machine-learning based crossing of mechanical characterisations of various cells in different states and their transcriptional profiles. Using this approach, they identify a core set of five genes that systematically vary together with the mechanical state of the cells, although not always in the same direction depending on the conditions. They show that the combined transcriptional changes in this gene set is strongly predictive of a change in the cell mechanical properties, in systems that were not used to identify the genes (a validation set). Finally, they experimentally after the expression level of one of these genes, CAV1, that codes for the caveolin 1 protein, and show that, in a variety of cellular systems and contexts, perturbations in the expression level of CAV1 also induce changes in cell mechanics, cells with lower CAV1 expression being generally softer.

Overall the approach seems accessible, sound and is well described. My personal expertize is not suited to judge its validity, novelty or relevance, so I do not make comments on that. The results it provides seem to have been thoroughly tested by the authors (using different types of mechanical characterisations of the cells) and to be robust in their predictive value. The authors also show convincingly that one of the genes they identified, CAV1, is not only correlated with the mechanical properties of cells, but also that changing its expression level affects cell mechanics. At this stage, the study appears mostly focused on the description and validation of the methodological approach, and it is hard to really understand what the results obtain really mean, the importance of the biological finding - what is this set of 5 genes doing in the context of cell mechanics? Is it really central, or is it just one of the set of knobs on which the cell plays - and it is identified by this method because it is systematically modulated but maybe, for any given context, it is not the dominant player - all these fundamental questions remain unanswered at this stage. On one hand, it means that the study might have identified an important novel module of genes in cell mechanics, but on the other hand, it also reveals that it is not yet easy to interpret the results provided by this type of novel approach.

Comments on revisions:

In their point-by-point answer, the authors did a great effort to provide pedagogical answers that clarified most of the points I had raised. They also did more analysis, some of which are included as supplementary data, and added a few sentences to the main text and discussion. As far as I am concerned, I see no particular issue with the revised article. I think it will be interesting both as a new type of approach in mechanobiology, and also as a motivation for more experimentally oriented labs to test the hypothesis proposed in the article and the 'module' they found.

---

## [Author Response]

The following is the authors’ response to the original reviews.

In summary, the changes made in the revision process include:

An addition of a paragraph in the result section that discusses the absolute values of measured Young’s moduli in the light of probing frequencies, accompanied by a new supplementary figure and a supplementary table that support that discussion

- Fig. S10. Absolute Young’s modulus values across the frequencies characteristic for the three measurement methods.

- Table S9. Operation parameters of the three methods used for characterizing the mechanical properties of cells.

Three new supplementary figures that display the expression matrices for the genes from the identified modules in carcinoma datasets used for validation:

- Fig. S4. Expression of identified target genes in the CCLE microarray dataset used for validation.

- Fig. S5. Expression of identified target genes in the CCLE RNA-Seq dataset used for validation.

- Fig. S6. Expression of identified target genes in the Genentech dataset used for validation.

An addition of a paragraph in the discussion section that discusses the intracellular origins of resistance to deformation and the dominance of actin cortex at low deformations.

- Refinement of the manuscript text and figures based on the specific feedback from the Reviewers.

Please see below for detailed responses to the Reviewers’ comments.

**Reviewer #1 (Public Review)**
In this work, Urbanska and colleagues use a machine-learning based crossing of mechanical characterisations of various cells in different states and their transcriptional profiles. Using this approach, they identify a core set of five genes that systematically vary together with the mechanical state of the cells, although not always in the same direction depending on the conditions. They show that the combined transcriptional changes in this gene set is strongly predictive of a change in the cell mechanical properties, in systems that were not used to identify the genes (a validation set). Finally, they experimentally after the expression level of one of these genes, CAV1, that codes for the caveolin 1 protein, and show that, in a variety of cellular systems and contexts, perturbations in the expression level of CAV1 also induce changes in cell mechanics, cells with lower CAV1 expression being generally softer.Overall the approach seems accessible, sound and is well described. My personal expertise is not suited to judge its validity, novelty or relevance, so I do not make comments on that. The results it provides seem to have been thoroughly tested by the authors (using different types of mechanical characterisations of the cells) and to be robust in their predictive value. The authors also show convincingly that one of the genes they identified, CAV1, is not only correlated with the mechanical properties of cells, but also that changing its expression level affects cell mechanics. At this stage, the study appears mostly focused on the description and validation of the methodological approach, and it is hard to really understand what the results obtain really mean, the importance of the biological finding - what is this set of 5 genes doing in the context of cell mechanics? Is it really central, or is it just one of the set of knobs on which the cell plays - and it is identified by this method because it is systematically modulated but maybe, for any given context, it is not the dominant player - all these fundamental questions remain unanswered at this stage. On one hand, it means that the study might have identified an important novel module of genes in cell mechanics, but on the other hand, it also reveals that it is not yet easy to interpret the results provided by this type of novel approach.

We thank the Reviewer #1 for the thoughtful evaluation of our manuscript. The primary goal of the manuscript was to present a demonstration of an unbiased approach for the identification of genes involved in the regulations of cell mechanics. The manuscript further provides a comprehensive computational validation of all genes from the identified network, and experimental validation of a selected gene, CAV1.

We agree that at the current stage, far-reaching conclusions about the biological meaning of the identified network cannot be made. We are, however, convinced that the identification of an apparently central player such as CAV1 across various cellular systems is per se meaningful, in particular since CAV1 modulation shows clear effects on the cell mechanical state in several cell types.

We anticipate that our findings will encourage more mechanistic studies in the future, investigating how these identified genes regulate mechanical properties and interact with each other. Notwithstanding, the identified genes (after testing in specific system of interest) can be readily used as genetic targets for modulating mechanical properties of cells. Access to such modifications is of huge relevance not only for performing further research on the functional consequence of cell mechanics changes (in particular in in-vivo systems where using chemical perturbations is not always possible), but also for the potential future implementation in modulating mechanical properties of the cells to prevent disease (for example to inhibit cancer metastasis or increase efficacy of cancer cell killing by cytotoxic T cells).

We have now added a following sentence in the first paragraph of discussion to acknowledge the open ends of our study:

“(...). Here we leveraged this opportunity by performing discriminative network analysis on transcriptomes associated with mechanical phenotype changes to elucidate a conserved module of five genes potentially involved in cell mechanical phenotype regulation. We provided evidence that the inferred conserved functional network module contains an ensemble of five genes that, in particular when combined in a unique combinatorial marker, are universal, specific and trustworthy markers of mechanical phenotype across the studied mouse and human systems. We further demonstrated on the example of a selected marker gene, CAV1, that its experimental up- and downregulation impacts the stiffness of the measured cells. This demonstrates that the level of CAV1 not only correlates with, but also is causative of mechanical phenotype change. The mechanistic insights into how precisely the identified genes are involved in regulating mechanical properties, how they interact with each other, and whether they are universal and dominant in various contexts all remain to be established in

future studies.”

**Reviewer #2 (Public Review)**
A key strength is the quantitative approaches all add rigor to what is being attempted. The approach with very different cell culture lines will in principle help identify constitutive genes that vary in a particular and predictable way. To my knowledge, one other study that should be cited posed a similar pan-tissue question using mass spectrometry proteomics instead of gene expression, and also identified a caveolae component (cavin-1, PTRF) that exhibited a trend with stiffness across all sampled tissues. The study focused instead on a nuclear lamina protein that was also perturbed in vitro and shown to follow the expected mechanical trend (Swift et al 2013).

We thank the Reviewer #2 for the positive evaluation of the breadth of the results and for pointing us to the relevant reference for the proteomic analysis related to tissue stiffness (Swift et al., 2013). This study, which focused primarily on the tissue-level mechanical properties, identifying PTRF, a caveolar component, which links to our observation of another caveolar component, CAV1, at the single-cell level.

We have now included the citation in the following paragraph of the discussion:

“To our knowledge, there are no prior studies that aim at identifying gene signatures associated with single-cell mechanical phenotype changes, in particular across different cell types. There are, however, several studies that investigated changes in expression upon exposure of specific cell types to mechanical stimuli such as compression (87, 88) or mechanical stretch (22, 80, 89), and one study that investigated difference in expression profiles between stiffer and softer cells sorted from the same population (90). Even though the studies concerned with response to mechanical stimuli answer a fundamentally different question (how gene expression changes upon exposure to external forces vs which genes are expressed in cells of different mechanical phenotype), we did observe some similarities in the identified genes. For example, in the differentially expressed genes identified in the lung epithelia exposed to compression (87), three genes from our module overlapped with the immediate response (CAV1, FHL2, TGLN) and four with the long-term one (CAV1, FHL2, TGLN, THBS1). We speculate that this substantial overlap is caused by the cells undergoing change in their stiffness during the response to compression (and concomitant unjamming transition). Another previous study explored the association between the stiffness of various tissues and their proteomes. Despite the focus on the tissue-scale rather than single-cell elasticity, the authors identified polymerase I and transcript release factor (PTRF, also known as cavin 1 and encoding for a structural component of the caveolae) as one of the proteins that scaled with tissue stiffness across samples (91).”

**Reviewer #3 (Public Review)**
In this work, Urbanska et al. link the mechanical phenotypes of human glioblastoma cell lines and murine iPSCs to their transcriptome, and using machine learning-based network analysis identify genes with putative roles in cell mechanics regulation. The authors identify 5 target genes whose transcription creates a combinatorial marker which can predict cell stiffness in human carcinoma and breast epithelium cell lines as well as in developing mouse neurons. For one of the target genes, caveolin1 (CAV1), the authors perform knockout, knockdown, overexpression and rescue experiments in human carcinoma and breast epithelium cell lines. They determine the cell stiffness via RT-DC, AFM indentation and AFM rheology and confirm that high CAV1 expression levels correlate with increased stiffness in those model systems. This work brings forward an interesting approach to identify novel genes in an unbiased manner, but surprisingly the authors validate caveolin 1, a target gene with known roles in cell mechanics regulation.I have two main concerns with the current version of this work:(1) The authors identify a network of 5 genes that can predict mechanics. What is the relationship between the 5 genes? If the authors aim to highlight the power of their approach by knockdown, knockout or over-expression of a single gene why choose CAV1 (which has an individual p-value of 0.16 in Fig S4)? To justify their choice, the authors claim that there is limited data supporting the direct impact of CAV1 on mechanical properties of cells but several studies have previously shown its role in for example zebrafish heart stiffness, where a knockout leads to higher stiffness (Grivas et al., Scientific Reports 2020), in cancer cells, where a knockdown leads to cell softening (Lin et al., Oncotarget 2015), or in endothelial cell, where a knockout leads to cell softening (Le Master et al., Scientific Reports 2022).

We thank the reviewer for their comments. First, we do acknowledge that studying the relationship between the five identified genes is an intriguing question and would be a natural extension of the currently presented work. It is, however, beyond the scope of presented manuscript, in which our primarily goal was to introduce a general pipeline for de novo identification of genes related to cell mechanics. We did add a following statement in the discussion (yellow highlight) to acknowledge the open ends of our study:

“The mechanical phenotype of cells is recognized as a hallmark of many physiological and pathological processes. Understanding how to control it is a necessary next step that will facilitate exploring the impact of cell mechanics perturbations on cell and tissue function (76).

The increasing availability of transcriptional profiles accompanying cell state changes has recently been complemented by the ease of screening for mechanical phenotypes of cells thanks to the advent of high-throughput microfluidic methods (77). This provides an opportunity for data-driven identification of genes associated with the mechanical cell phenotype change in a hypothesis-free manner. Here we leveraged this opportunity by performing discriminative network analysis on transcriptomes associated with mechanical phenotype changes to elucidate a conserved module of five genes potentially involved in cell mechanical phenotype regulation. We provided evidence that the inferred conserved functional network module contains an ensemble of five genes that, in particular when combined in a unique combinatorial marker, are universal, specific and trustworthy markers of mechanical phenotype across the studied mouse and human systems. We further demonstrated on the example of a selected marker gene, CAV1, that its experimental up- and downregulation impacts the stiffness of the measured cells. This demonstrates that the level of CAV1 not only correlates with, but also is causative of mechanical phenotype change. The mechanistic insights into how precisely the identified genes are involved in regulating mechanical properties, how they interact with each other, and whether they are universal and dominant in various contexts all remain to be established in future studies.”

Regarding the selection of CAV1 as the gene that we used for validation experiment; as mentioned in the introductory paragraph of the result section “Perturbing expression levels of CAV1 changes cells stiffness” (copied below), we were encouraged by the previous data already linking CAV1 with cell mechanics when selecting it as our first target. The relationship between CAV1 and cell mechanics regulation, however, is not very well established (of note, two of the latest manuscripts came out after the initial findings of our study).

Regarding the citations suggested by the reviewer: two are already included in the original manuscript (Lin et al., Oncotarget 2015 – Ref (63), Le Master –2022 Ref (67)), along with an additional one (Hsu et al 2018 (66)), and the third one (Grivas et al, 2020 (68)) is now also added to the manuscript. Though, we would like to highlight that even though Grivas et al state that the CAV1 KO cells are stiffer, the AFM indentation measurements were performed on the cardiac tissue, with a spherical tip of 30 μm radius and likely reflect primarily supracelluar, tissue-scale properties, as opposed to cell-scale measurements performed in our study (we used cultured cells which mostly lack the extracellular tissue structures, deformability cytometry was performed on dissociated cells and picks up on cell properties exclusively, and in case of AFM measurements a spherical tip with 5 μm radius was used).

“We decided to focus our attention on CAV1 as a potential target for modulating mechanical properties of cells, as it has previously been linked to processes intertwined with cell mechanics. In the context of mechanosensing, CAV1 is known to facilitate buffering of the membrane tension (45), play a role in β1-inegrin-dependent mechanotransduction (58) and modulate the mechanotransduction in response to substrate stiffness (59). CAV1 is also intimately linked with actin cytoskeleton — it was shown to be involved in cross-talk with Rho-signaling and actin cytoskeleton regulation (46, 60–62), filamin A-mediated interactions with actin filaments (63), and co-localization with peripheral actin (64). The evidence directly relating CAV1 levels with the mechanical properties of cells (47, 62, 65, 66) and tissues (66, 67) , is only beginning to emerge.”

Regarding the cited p-value of 0.16, we would like to clarify that it is the p-value associated with the coefficient of the crude linear regression model fitted to the data for illustrative purposes in Fig S4. This value only says that from the linear fit we cannot conclude much about the correlation of the level of Cav1 with the Young’s modulus change. Much more relevant parameters to look at are the AUC-ROC values and associated p-values reported in the Table 4 in the main text (see below), which show good performance of CAV1 in separating soft and stiff cell states.

The positive hypothesis I assumes that markers are discriminative of samples with stiff/soft mechanical phenotype regardless of the studied biological system, and CAV1 has a clear trend with the minimum AUC-ROC on 3 datasets of 0.78, even though the p-value is below the significance level. The positive hypothesis II assumes that markers are discriminative of samples with stiff/soft mechanical phenotype in carcinoma regardless of data source, and CAV1 has a clear significance because the minimum AUC-ROC on 3 datasets is 0.89 and the p-value is 0.02.

(2) The authors do not show how much does PC-Corr outperforms classical co-expression network analysis or an alternative gold standard. It is worth noting that PC-Corr was previously published by the same authors to infer phenotype-associated functional network modules from omics datasets (Ciucci et al., Scientific Reports 2017).

As pointed out by the Reviewer, PC-corr has been introduced and characterized in detail in a previous publication (Ciucci et al, 2017, Sci. Rep.), where it was compared against standard co-expression analysis (below reported as: p-value network) on molecules selected using univariate statistical analysis.

See the following fragment of Discussion in Ciucci et al, 2017:

“The PC-corr networks were always compared to P-value networks. The first strategical difference lies in the way features are selected: while the PC-corr adopts a multivariate approach, i.e. it uses a combination of features that are responsible for the sample discrimination, in the P-value network the discriminating features are singly selected (one by one) with each Mann-Whitney test (followed by Benjamini-Hochberg procedure). The second strategical difference lies in the generation of the correlation weights in the network. PC-corr combines in parallel and at the same time in a unique formula the discrimination power of the PC-loadings and the association power of the Pearson correlation, directly providing in output discriminative omic associations. These are generated using a robust (because we use as merging factor the minimum operator, which is a very penalizing operator) mathematical trade-off between two important factors: multivariate discriminative significance and correlation association. In addition, as mentioned above, the minimum operator works as an AND logical gate in a digital circuit, therefore in order to have a high link weight in the PCcorr network, both the discrimination (the PC-loadings) and the association (the Pearson correlations) of the nodes adjacent to the link should be simultaneously high. Instead, the Pvalue procedure begins with the pre-selection of the significant omic features and, only in a second separated step, computes the associations between these features. Therefore, in P-value networks, the interaction weights are the result neither of multivariate discriminative significance, nor of a discrimination/association interplay.”

Here we implement PC-corr for a particular application and do not see it as central to the message of the present manuscript to compare it with other available methods. We considered it much more relevant to focus on an in-silico validation on dataset not used during the PCcorr analysis (see Table 3 and 4 for details).

Altogether, the authors provide an interesting approach to identify novel genes associated with cell mechanics changes, but the current version does not fulfill such potential by focusing on a single gene with known roles in cell mechanics.

Our manuscript presents a demonstration of an overall approach for the identification of genes involved in the regulation of cell mechanics, and the perturbations performed on CAV1 have a demonstrative role (please also refer to the explanations of why we decided to perform the verification focused on CAV1 above). The fact that we identify CAV1, which has been implicated in regulating cell mechanics in a handful of studies, de novo and in an unbiased way speaks to the power of our approach. We do agree that investigation into the effect of manipulating the expression of the remaining genes from the identified network module, as well as into the mutual relationships between those genes and their covariance in perturbation experiments, constitutes a desirable follow-up on the presented results. It is, however, beyond the scope of the current manuscript. Regardless, the other genes identified can be readily tested in systems of interest and used as potential knobs for tuning mechanical properties on demand.

**Reviewer #1 (Recommendations For Authors)**
I am not a specialist of the bio-informatics methods used in this study, so I will not make any specific technical comments on them.In terms of mechanical characterisation of cells, the authors use well established methods and the fact that they systematically validate their findings with at least two independent methods (RT-DC and AFM for example) makes them very robust. So I have no concerns with this part. The experiments of perturbations of CAV 1 are also performed to the best standards and the results are clear, no concern on that.My main concerns are rather questions I was asking myself and could not answer when reading the article. Maybe the authors could find ways to clarify them - the discussion of their article is already very long and maybe it should not be lengthened to much. In my opinion, some of the points discussed are not really essential and rather redundant with other parts of the paper. This could be improved to give some space to clarify some of the points below:

We thank the Reviewer #1 for an overall positive evaluation of the manuscript as well as the points of criticism which we addressed in a point-by-point manner below.

(1) This might be a misunderstanding of the method on my side, but I was wondering whether it is possible to proceed through the same steps but choose other pairs of training datasets amongst the 5 systems available (there are 10 such pairs if I am not mistaken) and ask whether they always give the same set of 5 genes. And if not, are the other sets also then predictive, robust, etc. Or is it that there are 'better' pairs than others in this respect. Or the set of 5 genes is the only one that could be found amongst these 5 datasets - and then could it imply that it is the only group 'universal' group of predictive genes for cell mechanics (when applied to any other dataset comprising similar mechanical measures and expression profiles, for other cells, other conditions)?

I apologize in case this question is just the result of a basic misunderstanding of the method on my side. But I could not answer the question myself based on what is in the article and it seems to be important to understand the significance of the finding and the robustness of the method.

We thank the Reviewer for this question. To clarify: while in general it is possible to proceed through the same analysis steps choosing a different pair of datasets (see below for examples), we have purposefully chosen those two and not any other datasets because they encompassed the highest number of samples per condition in the RNAseq data (see Fig 4 and Table R1 below), originated from two different species and concerned least related tissues (the other option for mouse would be neural progenitors which in combination with the glioblastoma would likely result in focusing on genes expressed in neural tissues). This is briefly explained in the following fragment of the manuscript on Page 10:

“For the network construction, we chose two datasets that originate from different species, concern unrelated biological processes, and have a high number of samples included in the transcriptional analysis: human glioblastoma and murine iPSCs (Table 1).”

To further address the comment of the reviewer: there is indeed a total of 10 possible two-set combinations of datasets, 6 of those pairs are human-mouse combinations (highlighted in orange in Author response Table 1), 3 are human-human combinations (highlighted in blue), and 1 is mousemouse (marked in green).

**Author response table 1. sa2table1:** Possible two-set combinations of datasets. For each combination, the number of common genes is indicated. The number on the diagonal represents total number of transcripts in the individual datasets, n corresponds to the number of samples in the respective datasets. * include non-coding genes.

		human	mouse
		A. glioblastoma	B. carcinoma	C. MCF10A	D. IPSCs	E. dev neurons
**human**	A. glioblastoma	39400*(n = 27)	16260	31688	9452	12894
B. carcinoma		18821(n = 22)	16267	9410	12393
C. MCF10A			38508*(n = 6)	9598	12951
**mouse**	D. IPSCs				18118(n = 28)	10338
E. dev neurons					21110(n = 6)

To reiterate, we have chosen the combination of set A (glioblastoma) and set D (iPSCs) to choose datasets from different species and with highest sample number.

As for the other combinations of human-mouse datasets:

• set A & E lead to derivation of a conserved module, however as expected this module includes genes specific for neuronal tissues (such as brain & testis specific immunoglobulin IGSF11, or genes involved in neuronal development such as RFX4, SOX8)

**Author response image 1. sa2fig1:** 

• the remaining combinations (set B&D, B&E, C&D and C&E) do not lead to a derivation of a highly interconnected module

**Author response image 3. sa2fig3:** 

**Author response image 4. sa2fig4:** 

**Author response image 5. sa2fig5:** 

Finally, it would have also been possible to perform the combined PC-corr procedure on all 5 datasets. However, this would prevent us from doing validation using unknown datasets.

Hence, we decided to proceed with the 2 discovery and 4 validation datasets.

For the sake of completeness, we present below some of the networks obtained from the analysis performed on all 5 datasets (which intersect at 8059 genes).

**Author response image 6. sa2fig6:** 

The above network was created by calculating mean/minimum PC-corr among all five datasets and applying the threshold. The thresholding can be additionally restricted in that we:

a. constrain the directionality of the correlation between the genes (𝑠𝑔𝑛(𝑐)) to be the same among all or at least n datasets

b. constrain the directionality of the correlation between the cell stiffness and gene expression level (𝑠𝑔𝑛(𝑉)) for individual genes.

Some of the resulting networks for such restrictions are presented below.

**Author response image 7. sa2fig7:** 

**Author response image 8. sa2fig8:** 

Of note, some of the nodes from the original network presented in the paper (CAV1, FHL2, and IGFBP7) are preserved in the 5-set network (and highlighted with blue rims),

(2) The authors already use several types of mechanical characterisation of the cells, but there are even more of them, in particular, some that might not directly correspond to global cell stiffness but to other aspects, like traction forces, or cell cortex rheology, or cell volume or passage time trough constrictions (active or passive) - they might all be in a way or another related, but they are a priori independent measures. Would the authors anticipate finding very different 'universal modules' for these other mechanical properties, or again the same one? Is there a way to get at least a hint based on some published characterisations for the cells used in the study? Basically, the question is whether the gene set identified is specific for a precise type of mechanical property of the cell, or is more generally related to cell mechanics modulation - maybe, as suggested by the authors because it is a set of molecular knobs acting upstream of general mechanics effectors like YAP/TAZ or acto-myosin?

We thank the Reviewer for this comment. We would like to first note that in our study, we focused on single-cell mechanical phenotype understood as a response of the cells to deformation at a global (RT-DC) or semi-local (AFM indentation with 5-μm bead) level and comparatively low deformations (1-3 μm, see Table S9). There is of course a variety of other methods for measuring cell mechanics and mechanics-related features, such as traction force microscopy mentioned by the reviewer. Though, traction force microscopy probes how the cells apply forces and interact with their environment rather than the inherent mechanical properties of the cells themselves which were the main interest of our study.

Nevertheless, as mentioned in the discussion, we found some overlap with the genes identified in other mechanical contexts, for example in the context of mechanical stretching of cells:

“Furthermore, CAV1 is known to modulate the activation of transcriptional cofactor yesassociated protein, YAP, in response to changes in stiffness of cell substrate (60) and in the mechanical stretch-induced mesothelial to mesenchymal transition (74).”

Which suggests that the genes identified here may be more broadly related to mechanical aspects of cells.

Of note, we do have some insights connected to the changes of cell volume — one of the biophysical properties mentioned by the reviewer — from our experiments. For all measurements performed with RT-DC, we can also calculate cell volumes from 2D cell contours (see Author response images 9, 10, and 11). For most of the cases (all apart from MEF CAV1KO), the stiffer phenotype of the cells, associated with higher levels of CAV1, shows a higher volume.

**Author response image 9. sa2fig9:** Cell volumes for the divergent cell states in the five characterized biological systems. (**A**) Glioblastoma. (**B**) Carcinoma. (**C**) MCF10A. (**D**) iPSCs. (**E**) Developing neurons. Data corresponds to Figure 2. Cell volumes were estimated using Shape-Out 1.0.10 by rotation of the cell contours.

**Author response image 10. sa2fig10:** Cell volumes for CAV1 perturbation experiments. (**A**) CAV1 knock down performed in TGBC cells. (**B**) CAV1 overexpression in ECC4 and TGBC cells. Data corresponds to Figure 5. Cell volumes were estimated using Shape-Out 1.0.10 by rotation of the cell contours.

**Author response image 11. sa2fig11:** Cell volumes for WT and CAV1KO MEFs. Data corresponds to Figure S9. Cell volumes were estimated using Shape-Out 1.0.10 by rotation of the cell contours.

(3) The authors have already tested a large number of conditions in which perturbations of the level of expression of CAV1 correlates with changes in cell mechanics, but I was wondering whether it also has some direct explanatory value for the initial datasets used - for example for the glioblastoma cells from Figure 2, in the different media, would a knock-down of CAV1 prevent the increase in stiffness observed upon addition of serum, or for the carcinoma cells from different tissues treated with different compounds - if I understand well, the authors have tested a subset of these (ECC4 versus TGBC in figure 5) - how did they choose these and how general is it that the mechanical phenotype changes reported in Figure 2 are all mostly dependant on CAV1 expression level? I must say that the way the text is written and the results shown, it is hard to tell whether CAV1 is really having a dominant effect on cell mechanics in most of these contexts or only a partial effect. I hope I am being clear in my question - I am not questioning the conclusions of Figures 5 and 6, but asking whether the level of expression of CAV1, in the datasets reported in Figure 2, is the dominant explanatory feature for the differences in cell mechanics.

We thank reviewer for this comment and appreciate the value of the question about the generality and dominance of CAV1 in influencing cell mechanics.

On the computational side, we have addressed these issues by looking at the performance of CAV1 (among other identified genes) in classifying soft and stiff phenotypes across biological systems (positive hypothesis I), as well as across data of different type (sequencing vs microarray data) and origin (different research institutions) (positive hypothesis II). CAV1 showed strong classification performance (Table 4), suggesting it is a general marker of stiffness changes.

On the experimental side, we conducted the perturbation experiments in two systems of choice: two intestinal carcinoma cell lines (ECC4 and TGBC) and the MCF10A breast epithelial cell line. These choices were driven by ease of handling, accessibility, as well as (for MCF10A) connection with a former study (Taveres et al, 2017). While we observed correlations between CAV1 expression and cell mechanics in wide range of datasets, the precise role of CAV1 in each system may vary, and further perturbation experiments in specific systems could be performed to solidify the direct/dominant role of CAV1 in cell mechanics. We hypothesize that the suggested knockdown of CAV1 upon serum addition in glioblastoma cells could reduce or prevent the increase in stiffness observed, though this experiment has not been performed.

In conclusion, while the computational analysis gives us confidence that CAV1 is a good indicator of cell stiffness, we predict that it acts in concert with other genes and in specific context could be replaced by other changes. We suggest that the suitability of CAV1 for manipulation of the mechanical properties should be tested in each system of interested before use.

To highlight the fact that the relevance of CAV1 for modulating cell mechanics in specific systems of interest should be tested and the mechanistic insights into how CAV1 regulates cell mechanics are still missing, we have added the following sentence in the discussion:

“The mechanical phenotype of cells is recognized as a hallmark of many physiological and pathological processes. Understanding how to control it is a necessary next step that will facilitate exploring the impact of cell mechanics perturbations on cell and tissue function (76). The increasing availability of transcriptional profiles accompanying cell state changes has recently been complemented by the ease of screening for mechanical phenotypes of cells thanks to the advent of high-throughput microfluidic methods (77). This provides an opportunity for data-driven identification of genes associated with the mechanical cell phenotype change in a hypothesis-free manner. Here we leveraged this opportunity by performing discriminative network analysis on transcriptomes associated with mechanical phenotype changes to elucidate a conserved module of five genes potentially involved in cell mechanical phenotype regulation. We provided evidence that the inferred conserved functional network module contains an ensemble of five genes that, in particular when combined in a unique combinatorial marker, are universal, specific and trustworthy markers of mechanical phenotype across the studied mouse and human systems. We further demonstrated on the example of a selected marker gene, CAV1, that its experimental up- and downregulation impacts the stiffness of the measured cells. This demonstrates that the level of CAV1 not only correlates with, but also is causative of mechanical phenotype change. The mechanistic insights into how precisely the identified genes are involved in regulating mechanical properties, how they interact with each other, and whether they are universal and dominant in various contexts all remain to be established in future studies.”

(4) It would be nice that the authors try to more directly address, in their discussion, what is the biological meaning of the set of 5 genes that they found - is it really mostly a product of the methodology used, useful but with little specific relevance to any biology, or does it have a deeper meaning? Either at a system level, or at an evolutionary level.

We would like to highlight that our manuscript is focused on the method that we introduce to identify sets of genes involved in the regulation of cell mechanics. The first implementation included here is only the beginning of this line of work which, in the future, will include looking in detail at the biological meaning and the interconnectivity of the genes identified. Most likely, there is a deeper meaning of the identified module which could be revealed with a lot of dedicated future work. As it is a mere speculation at this point, we would like to refrain from going into more detail about it in the current manuscript. We provide below a few words of extended explanation and additional analysis that can shed light on the current limited knowledge of the connections between the genes and evolutionary preservation of the genes.

While it is difficult to prove at present, we do believe that the identified node of genes may have an actual biological meaning and is not a mere product of the used methodology. The PC-corr score used for applying the threshold and obtaining the gene network is high only if the Pearson’s correlation between the two genes is high, meaning that the high connected module of genes identified show corelated expression and is likely co-regulated. Additionally, we performed the GO Term analysis using DAVID to assess the connections between the genes (Figure S3). We have now performed an additional analysis using two orthogonal tools the functional protein association tool STRING and KEGG Mapper.

With STRING, we found a moderate connectivity using the five network nodes identified in our study, and many of the obtained connections were based on text mining and co-expression, rather than direct experimental evidence (Author response image 12A). A more connected network can be obtained by allowing STRING to introduce further nodes (Author response image 12B). Interestingly, some of the nodes included by STRING in the extended network are nodes identified with milder PCcorr thresholds in our study (such as CNN2 or IGFBP3, see Table S3).

With KEGG Mapper, we did not find an obvious pathway-based clustering of the genes from the module either. A maximum of two genes were assigned to one pathway and those included:

• focal adhesions (pathway hsa04510): CAV1 and THBS1

• cytoskeleton in muscle cells (pathway hsa04820): FHL2 and THBS1

• proteoglycans in cancer (pathway hsa05205): CAV1 and THBS1.

As for the BRITE hierarchy, following classification was found:

• membrane trafficking(hsa04131): CAV1, IGFBP7, TAGLN, THBS, with following subcategories:

- endocytosis / lipid raft mediated endocytosis/caveolin-mediated endocytosis:

CAV1

- endocytosis / phagocytosis / opsonins: THBS1

- endocytosis / others/ insulin-like growth factor-binding proteins: IGFBP7 o others / actin-binding proteins/others: TAGLN.

Taken together, all that analyzes (DAVID, STRING, KEGG) show that at present no direct relationship/single pathway can be found that integrates all the genes from the identified modules. Future experiments, including investigations of how other module nodes are affected when one of the genes is manipulated, will help to establish actual physical or regulatory interactions between the genes from our module.

To touch upon the evolutionary perspective, we provide an overview of occurrence of the genes from the identified module across the evolutionary tree. This overview shows that the five identified genes are preserved in phylum Chordata with quite high sequence similarity, and even more so within mammals (Author response image 13).

**Author response image 12. sa2fig12:** Visualisation of interactions between the nodes in the identified module using functional protein association networks tool STRING. (**A**) Connections obtained using multiple proteins search and entering the five network nodes. (**B**) Extended network that includes further genes to increase indirect connectivity. The genes are added automatically by STRING. Online version of STRING v12.0 was used with *Homo sapiens* as species of interest.

**Author response image 13. sa2fig13:** Co-occurrence of genes from the network module across the evolutionary tree. Mammals are indicated with the green frame, glires (include mouse), as well as primates (include human) are indicated with yellow frames. The view was generated using online version of STRING 12.0.

**Reviewer #2 (Recommendations For Authors)**
(1) The authors need to discuss the level of sensitivity of their mechanical measurements with RT-DC for changes to the membrane compared to changes in microtubules, nucleus, etc. The limited AFM measurements also seem membrane/cortex focused. For these and further reasons below, "universal" doesn't seem appropriate in the title or abstract, and should be deleted.

We thank the reviewer for this comment. Indeed, RT-DC is a technique that deforms the entire cell to a relatively low degree (inducing ca 17% mean strain, i.e. a deformation of approximately 2.5 µm on a cell with a 15 µm diameter, see Table S9 and Urbanska et al., Nat Methods 2020). Similarly, the AFM indentation experiments performed in this study (using a 5-µm diameter colloidal probe and 1 µm indentation) induce low strains, at which, according to current knowledge, the actin cortex dominates the measured deformations. However, other cellular components, including the membrane, microtubules, intermediate filaments, nucleus, other organelles, and cytoplasmic packing, can also contribute. We have reviewed these contributions in detail in a recent publication (Urbanska and Guck, 2024, Ann Rev Biophys., PMID 38382116). For a particular system, it is hard to speculate without further investigation which parts of the cell have a dominant effect on the measured deformability. We have added now a following paragraph in the discussion to include this information:

“The mechanical phenotype of single cells is a global readout of cell’s resistance to deformation that integrates contributions from all cellular components. The two techniques implemented for measuring cell mechanical in this study — RT-DC and AFM indentation using a spherical indenter with 5 µm radius — exert comparatively low strain on cells (< 3 µm, see Table S9), at which the actin cortex is believed to dominate the measured response. However, other cellular components, including the membrane, microtubules, intermediate filaments, nucleus, other organelles, and cytoplasmic packing, also contribute to the measured deformations (reviewed in detail in (79)) and, for a particular system, it is hard to speculate without further investigation which parts of the cell have a dominant effect on the measured deformability.”

The key strength of measuring the global mechanics is that such measurements are agnostic of the specific origin of the resistance to shape change. As such, the term “universal” could be seen as rather appropriate, as we are not testing specific contributions to cell mechanics, and we see the two methods used (RT-DC and AFM indentation) as representative when it comes to measuring global cell mechanics. And we highlighted many times throughout the text that we are measuring global single-cell mechanical phenotype.

Most importantly, however, we have used the term “universal” to capture that the genes are preserved across different systems and species, not in relation to the type of mechanical measurements performed and as such we would like to retain the term in the title.

(2) Fig.2 cartoons of tissues is a good idea to quickly illustrate the range of cell culture lines studied. However, it obligates the authors to examine the relevant primary cell types in singlecell RNAseq of human and/or mouse tissues (e.g. Tabula Muris). They need to show CAV1 is expressed in glioblastoma, iPSCs, etc and not a cell culture artifact. CAV1 and the other genes also need to be plotted with literature values of tissue stiffness.

We thank the reviewer for this the comment; however, we do believe that the cartoons in Figure 2 should assist the reader to readily understand whether cultured cells derived from the respective tissues were used (see cartoons representing dishes), or the cells directly isolated from the tissue were measured (this is the case for the developing neurons dataset).

We did, however, follow the suggestion of the reviewer to use available resources and checked the expression of genes from the identified network module across various tissues in mouse and human. We first used the Mouse Genome Informatics (MGI; https://www.informatics.jax.org/) to visualize the expression of the genes across organs and organ systems (Author response image 14) as well as across more specific tissue structures (Author response image 15). These two figures show that the five identified genes are expressed quite broadly in mouse. We next looked at the expression of the five genes in the scRNASeq dataset from Tabula Muris (Author response image 16). Here, the expression of respective genes seemed more restricted to specific cell clusters. Finally, we also collected the cross-tissue expression of the genes from our module in human tissues from Human Protein Atlas v23 at both mRNA (Author response image 17) and protein (Author response image 18) levels. CAV1, IGFBP7, and THBS1 showed low tissue specificity at mRNA level, FHL2 was enriched in heart muscle and ovary (the heart enrichment is also visible in Author response image 15 for mouse) and TAGLN in endometrium and intestine. Interestingly, the expression at the protein level (Author response image 18) did not seem to follow faithfully the mRNA levels (Author response image 17). Overall, we conclude that the identified genes are expressed quite broadly across mouse and human tissues.

**Author response image 14. sa2fig14:** Expression of genes from the identified module across various organ and organ systems in mouse. The expression matrices for organs (**A**) and organ systems (**B**) were generated using Tissue x Gene Matrix tool of Gene eXpression Database (https://www.informatics.jax.org/gxd/, accessed on 22nd September 2024). No pre-selection of stage (age) and assay type (includes RNA and protein-based assays) was applied. The colors in the grid (blues for expression detected and reds for expression not detected) get progressively darker when there are more supporting annotations. The darker colors do not denote higher or lower levels of expression, just more evidence.

**Author response image 15. sa2fig15:** Expression of genes from the identified module across various mouse tissue structures. The expression matrices for age-selected mouse marked as adult (**A**) or young individuals (collected ages labelled P42-84 / P w6-w12 / P m1.5-3.0) (**B**) are presented and were generated using RNASeq Heatmap tool of Gene eXpression Database (https://www.informatics.jax.org/gxd/, accessed on 2nd October 2024).

**Author response image 16. sa2fig16:** Expression of genes from the identified module across various cell types and organs in t-SNE embedding of *Tabula Muris* dataset. (**A**) t-SNE clustering color-coded by organ. (**B-F**) t-SNE clustering colorcoded for expression of CAV1 (**B**), IGFBP7 (**C**), FHL2 (**D**), TAGLN (**E**), and THBS1 (**F**). The plots were generated using FACS-collected cells data through the visualisation tool available at https://tabulamuris.sf.czbiohub.org/ (accessed on 22nd September 2024).

**Author response image 17. sa2fig17:** Expression of genes from the identified module at the mRNA level across various human tissues. (**A-E**) Expression levels of CAV1 (**A**), IGFBP7 (**B**), FHL2 (**C**), TAGLN (**D**), and THBS1 (**E**). The plots were generated using consensus dataset from Human Protein Atlas v23 https://www.proteinatlas.org/ (accessed on 22nd September 2024).

**Author response image 18. sa2fig18:** Protein levels of genes from the identified module across various human tissues. (**A-E**) Protein levels of CAV1 (**A**), IGFBP7 (**B**), FHL2 (**C**), TAGLN (**D**), and THBS1 (**E**). The plots were generated using Human Protein Atlas v23 https://www.proteinatlas.org/ (accessed on 22nd September 2024).

Regarding literature values and tissue stiffness, we would like to argue that cell stiffness is not equivalent to tissue stiffness, and we are interested in the former. Tissue stiffness is governed by a combination of cell mechanical properties, cell adhesions, packing and the extracellular matrix. There can be, in fact, mechanically distinct cell types (for example characterized by different metabolic state, malignancy level etc) within one tissue of given stiffness. Hence, we consider that testing for the correlation between tissue stiffness and expression of identified genes is not immediately relevant.

(3) Fig.5D,H show important time-dependent mechanics that need to be used to provide explanations of the differences in RT-DC (5B,F) and in standard AFM indentation expts (5C,G). In particular, it looks to me that RT-DC is a high-f/short-time measurement compared to the AFM indentation, and an additional Main or Supp Fig needs to somehow combine all of this data to clarify this issue.

We thank the reviewer for this comment. It is indeed the case, that cells typically display higher stiffness when probed at higher rates. We have now expanded on this aspect of the results and added a supplementary figure (Fig. S10) that illustrates the frequencies used in different methods and summarizes the apparent Young’s moduli values into one plot in a frequencyordered manner. Of note, we typically acquire RT-DC measurements at up to three flowrates, and the increase in measurement flow rates accompanying increase in flow rate also results in higher extracted apparent Young’s moduli (see Fig. S10 B,D). We have further added Table S9 that summarizes operating parameters of all three methods used for probing cell mechanics in this manuscript:

“The three techniques for characterizing mechanical properties of cells — RT-DC, AFM indentation and AFM microrheology — differ in several aspects (summarized in Table S9), most notably in the frequency at which the force is applied to cells during the measurements, with RT-DC operating at the highest frequency (~600 Hz), AFM microrheology at a range of frequencies in-between (3–200 Hz), and AFM indentation operating at lowest frequency (5 Hz) (see Table S9 and Figure S10A). Even though the apparent Young’s moduli obtained for TGBCS cells were consistently higher than those for ECC4 cells across all three methods, the absolute values measured for a given cell line varied depending on the methods: RT-DC measurements yielded higher apparent Young’s moduli compared to AFM indentation, while the apparent Young’s moduli derived from AFM microrheology measurements were frequency-dependent and fell between the other two methods (Fig. 5B–D, Fig. S10B). The observed increase in apparent Young’s modulus with probing frequency aligns with previous findings on cell stiffening with increased probing rates observed for both AFM indentation (68, 69) and microrheology assays (70–72).”

(4) The plots in Fig.S4 are important as main Figs, particularly given the cartoons of different tissues in Fig.1,2. However, positive correlations for a few genes (CAV1, IGFBP7, TAGLN) are most clear for the multiple lineages that are the same (stomach) or similar (gli, neural & pluri). The authors need to add green lines and pink lines in all plots to indicate the 'lineagespecific' correlations, and provide measures where possible. Some genes clearly don't show the same trends and should be discussed.

We thank reviewer for this comment. It is indeed an interesting observation (and worth highlighting by adding the fits to lineage-restricted data) that the relationship between relative change in Young’s modulus and the selected gene expression becomes steeper for samples from similar tissue contexts.

For the sake of keeping the main manuscript compact, we decided to keep Fig. S7 (formerly Fig. S4) in the supplement, however, we did add the linear fit to the glioblastoma dataset (pink line) and a fit to the related neural/embryonic datasets (gli, neural & pluri – purple line) as advised — see below.

We did not pool the stomach data since it is represented by a single point in the figure, aligning with how the data is presented in the main text—stomach adenocarcinoma cell lines (MKN1 and MKN45) are pooled in Fig. 1B (see below).

We have also amended the respective results section to emphasize that, in certain instances, the correlation between changes in mechanical phenotype and alterations in the expression of analyzed genes may be less pronounced:

“The relation between normalized apparent Young’s modulus change and fold-change in the expression of the target genes is presented in Fig. S7. The direction of changes in the expression levels between the soft and stiff cell states in the validation datasets was not always following the same direction (Fig. 4, C to F, Fig. S7). This suggests that the genes associated with cell mechanics may not have a monotonic relationship with cell stiffness, but rather are characterized by different expression regimes in which the expression change in opposite directions can have the same effect on cell stiffness. Additionally, in specific cases a relatively high change in Young’s modulus did not correspond to marked expression changes of a given gene — see for example low CAV1 changes observed in MCF10A PIK3CA mutant (Fig. S7A), or low IGFBP7 changes in intestine and lung carcinoma samples (Fig. S7C). This indicates that the importance of specific targets for the mechanical phenotype change may vary depending on the origin of the sample.”

(5) Table-1 neuro: Perhaps I missed the use of the AFM measurements, but these need to be included more clearly in the Results somewhere.

To clarify: there were no AFM measurements performed for the developing neurons (neuro) dataset, and it is not marked as such in Table 1. There are previously published AFM measurements for the iPSCs dataset (maybe that caused the confusion?), and we referred to them as such in the table by citing the source Urbanska et al (30) as opposed to the statement “this paper” (see the last column of Table 1). We did not consider it necessary to include these previously published data. We have added additional horizontal lines to the table that will hopefully help in the table readability.

**Reviewer #3 (For Authors)**
Major- I strongly encourage the authors to validate their approach with a gene for which mechanical data does not exist yet, or explore how the combination of the 5 identified genes is the novel regulator of cell mechanics.

We appreciate the reviewer’s insightful comment and agree that it would be highly interesting to validate further targets and perform combinatorial perturbations. However, it is not feasible at this point to expand the experimental data beyond the one already provided. We hope that in the future, the collective effort of the cell mechanics community will establish more genes that can be used for tuning of mechanical properties of cells.

- If this paper aims at highlighting the power of PC-Corr as a novel inference approach, the authors should compare its predictive power to that of classical co-expression network analysis or an alternative gold standard.

We thank the reviewer for the suggestion to compare the predictive power of PC-Corr with classical co-expression network analysis or an alternative gold standard. PC-corr has been introduced and characterized in detail in a previous publication (Ciucci et al, 2017, Sci. Rep.), where it was compared against standard co-expression analysis methods. Here we implement PC-corr for a particular application. Thus, we do not see it as central to the message of the present manuscript to compare it with other available methods again.

- The authors call their 5 identified genes "universal, trustworthy and specific". While they provide a great amount of data all is derived from human and mouse cell lines. I suggest toning this down.

We thank the reviewers for this comment. To clarify, the terms universal, trustworthy and specific are based on the specific hypotheses tested in the validation part of the manuscript, but we understand that it may cause confusion. We have now toned that the statement by adding “universal, trustworthy and specific across the studied mouse and human systems” in the following text fragments:

(1) Abstract

“(…) We validate in silico that the identified gene markers are universal, trustworthy and specific to the mechanical phenotype across the studied mouse and human systems, and demonstrate experimentally that a selected target, CAV1, changes the mechanical phenotype of cells accordingly when silenced or overexpressed. (...)”

(2) Last paragraph of the introduction

“(…) We then test the ability of each gene to classify cell states according to cell stiffness in silico on six further transcriptomic datasets and show that the individual genes, as well as their compression into a combinatorial marker, are universally, specifically and trustworthily associated with the mechanical phenotype across the studied mouse and human systems. (…)”

(3) First paragraph of the discussion

“We provided strong evidence that the inferred conserved functional network module contains an ensemble of five genes that, in particular when combined in a unique combinatorial marker, are universal, specific and trustworthy markers of mechanical phenotype across the studied mouse and human systems.”

Minor suggestions- The authors point out how genes that regulate mechanics often display non-monotonic relations with their mechanical outcome. Indeed, in Fig.4 developing neurons have lower CAV1 in the stiff group. Perturbing CAV1 expression in that model could show the nonmonotonic relation and strengthen their claim.

We thank reviewer for highlighting this important point. It would indeed be interesting to explore the changes in cell stiffness upon perturbation of CAV1 in a system that has a potential to show an opposing behavior. Unfortunately, we are unable to expand the experimental part of the manuscript at this time. We do hope that this point can be addressed in future research, either by our team or other researchers in the field.

- In their gene ontology enrichment assay, the authors claim that their results point towards reduced transcriptional activity and reduced growth/proliferation in stiff compared to soft cells. Proving this with a simple proliferation assay would be a nice addition to the paper.

This is a valuable suggestion that should be followed up on in detail in the future. To give a preliminary insight into this line of investigation, we have had a look at the cell count data for the CAV1 knock down experiments in TGBC cells. Since CAV1 is associated with the GO Term “negative regulation of proliferation/transcription” (high CAV1 – low proliferation), we would expect that lowering the levels of CAV1 results in increased proliferation and higher cell counts at the end of experiment (3 days post transfection). As illustrated in Author response image 19 below, the cell counts were higher for the samples treated with CAV1 siRNAs, though, not in a statistically significant way. Interestingly, the magnitude of the effect partially mirrored the trends observed for the cell stiffness (Figure 5F).

**Author response image 19. sa2fig19:** The impact of CAV1 knock down on cell counts in TGBC cells. (**A**) Absolute cell counts per condition in a 6-well format. Cell counts were performed when harvesting for RT-DC measurements using an automated cell counter (Countess II, Thermo Fisher Scientific). (**B**) The event rates observed during the RT-DC measurements. The harvested cells are resuspended in a specific volume of measuring buffer standardized per experiment (50-100 μl); thus, the event rates reflect the absolute cell numbers in the respective samples. Horizontal lines delineate medians with mean absolute deviation (MAD) as error, datapoints represent individual measurement replicates, with symbols corresponding to matching measurement days. Statistical analysis was performed using two sample two-sided Wilcoxon rank sum test.

Methods- The AFM indentation experiments are performed with a very soft cantilever at very high speeds. Why? Also, please mention whether the complete AFM curve was fitted with the Hertz/Sneddon model or only a certain area around the contact point.

We thank the reviewer for this comment. However, we believe that the spring constants and indentation speeds used in our study are typical for measurements of cells and not a cause of concern.

For the indentation experiments, we used Arrow-TL1 cantilevers (nominal spring constant k = 0.035-0.045 N m^−1^, Nanoworld, Switzerland) which are used routinely for cell indentation (with over 200 search results on Google Scholar using the term: "Arrow-TL1"+"cell", and several former publications from our group, including Munder et al 2016, Tavares et al 2017, Urbanska et al 2017, Taubenberger et al 2019, Abuhattum et al 2022, among others). Additionally, cantilevers with the spring constants as low as 0.01 N m−1 can be used for cell measurements (Radmacher 2002, Thomas et al, 2013).

The indentation speed of 5 µm s^−1^ is not unusually high and does not result in significant hydrodynamic drag.

For the microrheology experiments, we used slightly stiffer and shorter (100/200 µm compared to 500 µm for Arrow-TL1) cantilevers: PNP-TR-TL (nominal spring constant k = 0.08 Nm^−1^, Nanoworld, Switzerland). The measurement frequencies of 3-200 Hz correspond to movements slightly faster than 5 µm s^−1^, but cells were indented only to 100 nm, and the data were corrected for the hydrodynamic drag (see equation (8) in Methods section).

**Author response image 20. sa2fig20:** 

Exemplary indentation curve obtained using arrow-TL1 decorated with a 5-µm sphere on a ECC4 cell. The shown plot is exported directly from JPK Data Processing software. The area shaded in grey is the area used for fitting the Sneddon model.

In the indentation experiments, the curves were fitted to a maximal indentation of 1.5 μm (rarely exceeded, see Author response image 20). We have now added this information to the methods section:

- Could the authors include the dataset wt #1 in Fig 4D? Does it display the same trend?

We thank the reviewer for this comment. To clarify: in the MCF10A dataset (GEO: GSE69822) there are exactly three replicates of each wt (wild type) and ki (knock-in, referring to the H1047R mutation in the PIK3CA) samples. The numbering wt#2, wt#3, wt#4 originated from the short names that were used in the working files containing non-averaged RPKM (possibly to three different measurement replicates that may have not been exactly paired with the ki samples). We have now renamed the samples as wt#1, wt#2 and wt#3 to avoid the confusion. This naming also reflects better the sample description as deposited in the GSE69822 dataset (see Author response table 2).

**Author response table 2. sa2table2:** 

Short Name	Sample No	Sample Description	Displayed Name (old)	Displayed Name (new)
wt_2_0	GSM1709515	MCF10a WT_t=0_replicate1condition: no EGF +DMSO	wt #2	wt #1
wt_3b_0	GSM1709516	MCF10a WT_t=0_replicate2condition: no EGF +DMSO	wt #3	wt #2
wt_4_0	GSM1709517	MCF10a WT_t=0_replicate3condition: no EGF +DMSO	wt #4	wt #3
ki_1_0	GSM1709554	MCF10a PIK3CAH1047R_t=0_replicate1condition: no EGF +DMSO	ki#1	ki#1
ki_2_0	GSM1709555	MCF10a PIK3CAH1047R_t=0_replicate2condition: no EGF +DMSO	ki#2	ki#2
ki_3_0	GSM1709556	MCF10a PIK3CAH1047R_t=0_replicate3condition: no EGF +DMSO	ki#3	ki#3

- Reference (3) is an opinion article with the last author as the sole author. It is used twice as a self-standing reference, which is confusing, as it suggests there is previous experimental evidence.

We thank the reviewer for pointing this out and agree that it may not be appropriate to cite the article (Guck 2019 Biophysical Reviews, formerly Reference (3), currently Reference (76)) in all instances. The references to this opinion article have now been removed from the introduction:

“The extent to which cells can be deformed by external loads is determined by their mechanical properties, such as cell stiffness. Since the mechanical phenotype of cells has been shown to reflect functional cell changes, it is now well established as a sensitive label-free biophysical marker of cell state in health and disease (1-2).”

“Alternatively, the problem can be reverse-engineered, in that omics datasets for systems with known mechanical phenotype changes are used for prediction of genes involved in the regulation of mechanical phenotype in a mechanomics approach.”

But has been kept in the discussion:

“The mechanical phenotype of cells is recognized as a hallmark of many physiological and pathological processes. Understanding how to control it is a necessary next step that will facilitate exploring the impact of cell mechanics perturbations on cell and tissue function

(76).”.

This reference seems appropriate to us as it expands on the point that our ability to control cell mechanics will enable the exploration of its impact on cell and tissue function, which is central to the discussion of the current manuscript.

-The authors should mention what PC-corr means. Principle component correlation? Pearson's coefficient correlation?

PC-corr is a combination of loadings from the principal component (PC) analysis and Pearson’s correlation for each gene pair. We have aimed at conveying this in the “Discriminative network analysis on prediction datasets” result section. We have now added and extra sentence at the first appearance of PC-corr to clarify that for the readers from the start:

“After characterizing the mechanical phenotype of the cell states, we set out to use the accompanying transcriptomic data to elucidate genes associated with the mechanical phenotype changes across the different model systems. To this end, we utilized a method for inferring phenotype-associated functional network modules from omics datasets termed PCCorr (28), that relies on combining loadings obtained from the principal component (PC) analysis and Pearson’s correlation (Corr) for every pair of genes. PC-Corr was performed individually on two prediction datasets, and the obtained results were overlayed to derive a conserved network module. Owing to the combination of the Pearson’s correlation coefficient and the discriminative information included in the PC loadings, the PC-corr analysis does not only consider gene co-expression — as is the case for classical co-expression network analysis — but also incorporates the relative relevance of each feature for discriminating between two or more conditions; in our case, the conditions representing soft and stiff phenotypes. The overlaying of the results from two different datasets allows for a multi-view analysis (utilizing multiple sets of features) and effectively merges the information from two different biological systems.”

- The formatting of Table 1 is confusing. Horizontal lines should be added to make it clear to the reader which datasets are human and which mouse as well as which accession numbers belong to the carcinomas.

Horizontal lines have now been added to improve the readability of Table 1. We hope that makes the table easier to follow and satisfies the request. We assume that further modifications to the table appearance may occur during publishing process in accordance with the publisher’s guidelines.

- In many figures, data points are shown in different shapes without an explanation of what the shapes represent.

We thank the reviewer for this comment and apologize for not adding this information earlier. We have added explanations of the symbols to captions of Figures 2, 3, 5, and 6 in the main text:

“Fig. 2. Mechanical properties of divergent cell states in five biological systems. Schematic overviews of the systems used in our study, alongside with the cell stiffness of individual cell states parametrized by Young’s moduli E. (…) Statistical analysis was performed using generalized linear mixed effects model. The symbol shapes represent measurements of cell lines derived from three different patients (A), matched experimental replicates (C), two different reprogramming series (D), and four different cell isolations (E). Data presented in (A) and (D) were previously published in ref (29) and (30), respectively.”

“Fig. 3. Identification of putative targets involved in cell mechanics regulation. (A) Glioblastoma and iPSC transcriptomes used for the target prediction intersect at 9,452 genes. (B, C) PCA separation along two first principal components of the mechanically distinct cell states in the glioblastoma (B) and iPSC (C) datasets. The analysis was performed using the gene expression data from the intersection presented in (A). The symbol shapes in (B) represent cell lines derived from three different patients. (…)”

“Fig. 5. Perturbing levels of CAV1 affects the mechanical phenotype of intestine carcinoma cells. (…) In (E), (F), (I), and (J), the symbol shapes represent experiment replicates.”

“Fig. 6. Perturbations of CAV1 levels in MCF10A-ER-Src cells result in cell stiffness changes. (…) Statistical analysis was performed using a two-sided Wilcoxon rank sum test. In (B), (D), and (E), the symbol shapes represent experiment replicates.”

As well as to Figures S2, S9, and S11 in the supplementary material (in Figure S2, the symbol explanation was added to the legends in the figure panels as well):

“Fig. S2. Plots of area vs deformation for different cell states in the characterized systems. Panels correspond to the following systems: (A) glioblastoma, (B) carcinoma, (C) non-tumorigenic breast epithelia MCF10A, (D) induced pluripotent stem cells (iPSCs), and (E) developing neurons. 95%- and 50% density contours of data pooled from all measurements of given cell state are indicated by shaded areas and continuous lines, respectively. Datapoints indicate medians of individual measurements. The symbol shapes represent cell lines derived from three different patients (A), two different reprogramming series (D), and four different cell isolations (E), as indicated in the respective panels. (…).”

“Fig. S9. CAV1 knock-out mouse embryonic fibroblasts (CAV1KO) have lower stiffness compared to the wild type cells (WT). (…) (C) Apparent Young’s modulus values estimated for WT and CAV1KO cells using areadeformation data in (B). The symbol shapes represent experimental replicates. (…)”

“Fig. S11. Plots of area vs deformation from RT-DC measurements of cells with perturbed CAV1 levels. Panels correspond to the following experiments: (A and B) CAV1 knock-down in TGBC cells using esiRNA (A) and ONTarget siRNA (B), (C and D) transient CAV1 overexpression in ECC4 cells (C) and TGBC cells (D). Datapoints indicate medians of individual measurement replicates. The isoelasticity lines in the background (gray) indicate regions of of same apparent Young’s moduli. The symbol shapes represent experimental replicates.”

- In Figure 2, the difference in stiffness appears bigger than it actually is because the y-axes are not starting at 0.

While we acknowledge that starting the y-axes at a value other than 0 is generally not ideal, we chose this approach to better display data variability and minimize empty space in the plots.

A similar effect can be achieved with logarithmic scaling, which is a common practice (see Author response image 21 for visualization). We believe our choice of axes cut-off enhances the interpretability of the data without misleading the viewer.

**Author response image 21. sa2fig21:** Visualization of different axis scaling strategies applied to the five datasets presented in Figure 2 of the manuscript.

Of note, apparent Young’s moduli obtained from RT-DC measurements typically span 0.5-3.0 kPa (see Figure 2.3 from Urbanska et al 2021, PhD thesis). Differences between treatments rarely exceed a few hundred pascals. For example, in an siRNA screen of mitotic cell mechanics regulators in *Drosophila* cells (Kc167), the strongest hits (e.g., Rho1, Rok, dia) showed changes in stiffness of 100-150 Pa (see Supplementary Figure 11 from Rosendahl, Plak et al 2018, Nature Methods 15(5): 355-358).

- In Figure 3, I don't personally see the benefit of showing different cut-offs for PC-corr. In the end, the paper focuses on the 5 genes in the pentagram. I think only showing one of the cutoffs and better explaining why those target genes were picked would be sufficient and make it clearer for the reader.

We believe it is beneficial to show the extended networks for a few reasons. First, it demonstrates how the selected targets connect to the broader panel of the genes, and that the selected module is indeed much more interconnected than other nodes. Secondly, the chosen PC-corr cut-off is somewhat arbitrary and it may be interesting to look through the genes from the extended network as well, as they are likely also important for regulating cell mechanics. This broader view may help readers identify familiar genes and recognizing the connections to relevant signaling networks and processes of interest.

- In Figure 4C, I suggest explaining why the FANTOM5 and not another dataset was used for the visualization here and mentioning whether the other datasets were similar.

In Figure 4C, we have chosen to present data corresponding to FANTOM5, because that was the only carcinoma dataset in which all the cell lines tested mechanically are presented. We have now added this information to the caption of Figure 4. Additionally, the clustergrams corresponding to the remaining carcinoma datasets (CCLE RNASeq, Genetech) are presented in supplementary figures S4-S6.

“The target genes show clear differences in expression levels between the soft and stiff cell states and provide for clustering of the samples corresponding to different cell stiffnesses in both prediction and validation datasets (Fig. 4, Figs. S4-S6).”

Typos

We would like to thank the Reviewer#3 for their detailed comments on the typos and details listed below. This is much appreciated as it improved the quality of our manuscript.

- In the first paragraph of the results section the 'and' should be removed from this sentence: Each dataset encompasses two or more cell states characterized by a distinct mechanical phenotype, and for which transcriptomic data is available.

The sentence has been corrected and now reads:

“Each dataset encompasses two or more cell states characterized by a distinct mechanical phenotype, and for which transcriptomic data is available.”

- In the methods in the MCF10A PIK3CA cell lines part, it says cell liens instead of cell lines.

The sentence has been corrected and now reads:

“The wt cells were additionally supplemented with 10 ng ml^−1^ EGF (E9644, Sigma-Aldrich), while mutant cell lienes were maintained without EGF.”

- In the legend of Figure 6 "accession number: GSE17941, data previously published in” the reference is missing.

The reference has been added.

- In the legend of Figure 5 "(E) Verification of CAV1 knock-down in TGBC cells using two knock-down system" 'a' between using and two is missing.

The legend has been corrected (no ‘a’ is missing, but it should say systems (plural)):

- In Figure 5B one horizontal line is missing.

The Figure 5B has been corrected accordingly.

- Terms such as de novo or in silico should be written in cursive.

We thank the Reviewer for this comment; however, we believe that in the style used by eLife, common Latin expressions such as de novo or in vitro are used in regular font.

- In the heading of Table 4 "The results presented in this table can be reproducible using the code and data available under the GitHub link reported in the methods section." It should say reproduced instead of reproducible.

Yes, indeed. It has been corrected.

- The citation of reference 20 contains several author names multiple times.

Indeed, it has been fixed now:

- In Figure S2 there is a vertical line in the zeros of the y axis labels.

I am not sure if there was some rendering issue, but we did not see a vertical line in the zeros of the y axis label in Figure S2.

- The Text in Figure S4 is too small.

We thank the reviewer for pointing this out. We have now revised Figure S7 (formerly Figure S4) to increase the text size, ensuring better readability. (It has also been updated to include additional fits as requested by Reviewer #2).

- In Table 3 "positive hypothesis II markers are discriminative of samples with stiff/soft independent of data source" the words 'mechanical phenotype' are missing.

The column headings in Table 3 have now been updated accordingly.

- In Table S3 explain in the table headline what vi1, vi2 and v are. I assume the loading for PC1, the loading for PC2 and the average of the previous two values. But it should be mentioned somewhere.

The caption of table S3 has been updated to explain the meaning of vi1, vi2 and v.